Open Access EGU

# The site-specified primary calibration conditions for the Brewer spectrophotometer

Xiaoyi Zhao[1], Vitali Fioletov[1], Alberto Redondas[2,3], Julian Gröbner[4], Luca Egli[4], Franz Zeilinger[4], Javier López-Solano[5,2,3], Alberto Berjon Arroyo[5,2,3], James Kerr[6], Eliane Maillard Barras[7], Herman Smit[8], Michael Brohart[1], Reno Sit[1], Akira Ogyu[1], Ihab Abboud[1], and Sum Chi Lee[1]

[1]Air Quality Research Division, Environment and Climate Change Canada, Toronto, M3H 5T4, Canada
[2]Agencia Estatal de Meteorología, Izaña Atmospheric Research Center, Tenerife, Spain
[3]Regional Brewer Calibration Center for Europe, Izaña Atmospheric Research Center, Tenerife, Spain
[4]Physikalisch-Meteorologisches Observatorium Davos, World Radiation Center, Switzerland
[5]TRAGSATEC, Madrid, Spain
[6]Retired senior research scientist of Environment and Climate Change Canada
[7]Federal Office of Meteorology and Climatology, MeteoSwiss, 1530 Payerne, Switzerland
[8]Forschungszentrum Jülich GmbH - Institute of Energy and Climate Research, Germany

*Correspondence to*: Xiaoyi Zhao (xiaoyi.zhao@ec.gc.ca)

**Abstract.** The Brewer ozone spectrophotometer (the Brewer) is one of the World Meteorological Organization (WMO) Global Atmosphere Watch (GAW) standard ozone monitoring instruments since the 1980s. The entire global Brewer ozone monitoring network is operated and maintained via a hierarchical calibration chain, which started from world reference instruments that are independently calibrated via the primary calibration method (PCM) at a premium site (National Oceanic and Atmospheric Administration's (NOAA) Mauna Loa Observatory, Hawaii). These world reference instruments have been maintained by Environment and Climate Change Canada (ECCC) in Toronto for the last four decades. Their calibration is transferred to the travelling standard instrument and then to network (field) Brewer instruments at their monitoring sites (all via the calibration transfer method; CTM). Thus, the measurement accuracy for the entire global network is dependent on the calibration of world reference instruments. In 2003, to coordinate regional calibration needs, the Regional Brewer Calibration Center-Europe (RBCC-E) was formed in Izaña, Spain. From that point, RBCC-E began calibrating regional references also via PCM, instead of CTM. The equivalency and consistency of world and regional references are then assured during international calibration campaigns. In practice, these two calibration methods have different physical requirements, e.g., the PCM requires a stable short-term ozone field, while CTM would benefit from larger changes in slant ozone conditions for the calibration periods. This difference dictates that the PCM can only be implemented on Brewers at certain sites and even in certain months of the year. This work is the first effort to use long-term observation records from 11 Brewers at four sites to reveal the challenges in performing PCM. By utilizing a new calibration simulation model and reanalysis ozone data, this work also quantifies uncertainties in the PCM due to short-term ozone variability. The results are validated by real-world observations and used to provide scientific advice on where and when the PCM can be performed and how many days of

observations are needed to achieve the calibration goal (i.e., ensure the calibration uncertainty is within a determined criterion, i.e., $\leq$5 R6 unit; R6 is a measurement derived double ratio in the actual Brewer processing algorithm). This work also suggests that even if the PCM cannot be used to deliver final calibration results for mid- or high-latitude sites, the statistics of the long-term PCM fitting results can still provide key information for field Brewers as stability indicators (which would provide performance monitoring and data quality assurance).

## 1 Introduction

As the major absorber of solar ultraviolet radiation in the 200–330 nm spectral band, ozone ($O_3$) is an important trace gas that influences the stratospheric and tropospheric dynamics and chemistry. Observations of atmospheric total column ozone (TCO) date back to the 1920s (Dobson, 1968), while the real concerns about the stratospheric ozone layer depletion were raised in the late 1970s to early 1980s. The discovery of the Antarctic ozone hole in 1985, findings of the vulnerability of stratospheric ozone to anthropogenic activities, global ozone decrease, and the consequential increase of ultraviolet-B (UV-B) radiation at the earth's surface resulted in stratospheric ozone research and monitoring to become urgent and important (Farman et al., 1985; Solomon et al., 1986; Stolarski et al., 1986; Ramaswamy et al., 1992). Since the late 1950s, Environment and Climate Change Canada (ECCC) has operated the Canadian ozone monitoring network that includes stations at Toronto, Edmonton, Churchill, Resolute, and Goose Bay, originally equipped with Dobson spectrophotometers (Dobson, 1931, 1968). To increase the quality and frequency and to automate ozone observations, Alan Brewer proposed a replacement to the Dobson instrument (Brewer, 1973) and the automated version of the Brewer instrument was developed in the early 1980s at ECCC (Brewer, 1973; Kerr et al., 1981, 1985b). Currently, with more than 240 Brewer instruments manufactured and deployed worldwide within the World Meteorological Organization (WMO) Global Atmosphere Watch (GAW) Programme, the Brewer network is the backbone of global ground-based ozone monitoring networks and plays an important role in ozone monitoring and trend detection (e.g., Fioletov et al., 1999; Staehelin et al., 2001; Weber et al., 2022), in satellite and model validation (e.g., Fioletov et al., 2006; Labow et al., 2013; Garane et al., 2019), as well as in assisting in the development of new ozone monitoring techniques (e.g., Zhao et al., 2016; Egli et al., 2022).

At the beginning of the Canadian Brewer Spectrophotometer Network, it was decided to follow a hierarchical calibration chain to maintain measurement stability and network consistency. Three Brewers were selected to form the world Brewer reference triad in Toronto (Kerr et al., 1985a) and receive their extraterrestrial calibration constant (ETC) via the independent primary calibration method (PCM; via Langley plot techniques) at the National Oceanic and Atmospheric Administration's Mauna Loa Observatory (MLO), Hawaii, U.S. every 3–8 years (Fioletov et al., 2005; Zhao et al., 2021). In Toronto, these world reference instruments were then used to transfer their calibration results to a travelling standard instrument via the calibration transfer method (CTM) (Kerr, 2010; Redondas et al., 2018a). Finally, the travelling standard instrument visits individual field

instruments at their observation sites (or during their participation in intercomparison campaigns) and transfers the calibration also via CTM. Thus, the accuracy of the entire global Brewer ozone monitoring network depends on the calibration of the world reference instruments. The Brewer reference triad (BrT), consisting of three Mark II instruments, was used instead of just one instrument to assure that any drifts of one instrument could be detected from the other two (Fioletov et al., 2005). Beginning in 2011, a double Brewer reference triad (BrT-D; consists of three Mark III instruments) started their service in Toronto (Zhao et al., 2021).

Besides the world reference triads, there are two other Brewer triads in operation in Europe, the Swiss triad and the Regional Brewer Calibration Center Europe (RBCC-E) triad. The Swiss triad, formed in 1998, is operated in Davos, Switzerland (relocated from Arosa between 2018 and 2021) by the PMOD/WRC for the Swiss Federal Office of Meteorology and Climatology (Meteo Swiss) (Staehelin et al., 1998; Stübi et al., 2017a). The RBCC-E triad, formed in 2003, is operated in Izaña, Spain, by the State Meteorological Agency of Spain (AEMET) (León-Luis et al., 2018) to facilitate the regional calibration needs. Using the same procedure as world reference instruments, these regional reference instruments (RBCC-E triad) are also calibrated via the PCM, but at their home site in Izaña. As a result, the consistency between the world and regional references must be assured during regular intercomparison campaigns (via comparing regional references with a travelling standard or world reference) (Redondas et al., 2018a).

Theoretically, the world and regional references, both independently calibrated at different sites, should have negligible differences in calibration results (e.g., accuracy). However, in practice, the AEMET and ECCC team uses different software, fitting air mass factor ranges, and data quality filters in their calibration work. More importantly, Langley plot techniques are based on an assumption that the ozone field is stable during each calibration session (i.e., half-day) and the observations are of good quality (e.g., observed spectral signals should not be "contaminated" by the presence of clouds or heavy aerosol scattering). In reality, these conditions at different locations may differ significantly in the calibration work. An alternate approach has been demonstrated by Egli et al. (2022) using a spectroradiometer calibrated in the laboratory with SI-traceable radiation standards to retrieve the atmospheric TCO from direct spectral solar irradiance measurements without requiring an in-situ based Langley calibration. As a result of the Langley calibrations, different calibration settings and site conditions could both have a clear impact on the final calibration results. Fortunately, from our analysis, the different calibration procedures (for Langley plot techniques) used by the AEMET and ECCC teams would have a negligible impact (more details will be published in an upcoming paper).

Meanwhile, the site conditions during the PCM calibrations, especially short-term ozone stability (i.e., TCO changes in half-day), could have a significant impact on the Langley plots and the PCM calibration results. It is worth noting that Kerr (2010) discussed the reasons for selecting MLO as the site to perform primary calibration, including its stable short-term ozone field

and high-altitude (less surface ozone pollution). The author provided also the number of days needed to accomplish high-quality calibration work (about 10 days in MLO) (Kerr, 2010). However, no indication of these conditions for other sites is given, nor any quantified comparisons between different sites are made. To further the effort, this study is focused on quantifying the site selection impact on the PCM using long-term observations from four sites (Arosa/Davos, Izaña, Toronto and MLO) with 11 Brewers total and a novel ETC simulation model. This work is aimed at obtaining answers to the following scientific questions — (1) why Brewer primary calibration work can only be performed at certain sites (e.g., Izaña and MLO) and (2) what is needed to assure the equivalence of calibration quality from different sites (e.g., how many days of observations are needed from different sites to achieve expected high-quality calibration results).

The paper is organized as follows: Section 2 introduces the instruments, reanalysis data, and sites. In Section 3, the primary calibration method, the ETC simulation model, and the validation and application work are described. In Section 4, other site-specified conditions, included to estimate the calibration needs and reveal different requirements, are described. Section 5 provides optimized primary calibration strategies that were applied by the AEMET and ECCC teams, and also how the findings in this work can assist in field Brewer operations, monitoring, and data assurance work. Conclusions are given in Section 6.

## 2 Instruments, reanalysis data, and sites

### 2.1 Brewer Spectrometer

The Brewer spectrophotometer is a modified Ebert grating spectrometer that was designed to measure almost simultaneously the intensity of radiation at six UV channels (nominal wavelength at 303.2, 306.3, 310.1, 313.5, 316.8, and 320.1 nm). The first channel is almost exclusively used for wavelength calibration. The four longer wavelengths are used for the TCO (Ω) retrieval via the following equation:

$$F + \Delta\beta \cdot m = F_0 - \Delta\alpha \cdot \Omega \cdot \mu \qquad (1)$$

where, $m$ and $\mu$ are the enhancement factors for the slant-to-vertical path length of the direct radiation for air and the ozone layer respectively (also known as the air mass factors). $F$, $\Delta\alpha$, and Δβ are the linear combinations of the logarithms of the measured intensity, the effective ozone absorption and the Rayleigh scattering coefficients, respectively. For example, $F = \log_{10}(I_3) - 0.5\log_{10}(I_4) - 2.2\log_{10}(I_5) + 1.7\log_{10}(I_6)$, where $I_3$ to $I_6$ are the photon count rates at the last four longer wavelength channels (Kerr et al., 1985b). $F_0$ is the instrument response if there were no atmosphere between the instrument and the sun, it is also known as the ETC. Here, Δβ, $m$, and $\mu$ are pre-calculated, and are not instrument-dependent. $F_0$ and $\Delta\alpha$ (calibration constants) are unique for each instrument and depend on the exact wavelengths and band passes of the slits of each

instrument. The details of the standard Brewer ozone retrieval algorithm can be found in Kerr (2010) and the references therein. In practice, $\Delta\alpha$ is determined via laboratory calibration (Gröbner et al., 1998; Redondas et al., 2018b), $F_0$ can be determined via two different field calibration approaches (i.e., PCM and CTM). After receiving these calibration constants, Ω is then readily calculated for each field observation (i.e., $F$). Total column ozone values in this study are given in Dobson Unit (DU; 1 DU = $2.6870 \times 10^{16}$ molec $cm^{-2}$).

There are several model versions of the Brewer instrument. Two Mark I prototype instruments were tested and operated briefly in the late 1970s (Kerr et al., 1981). The first production version (Mark II) was introduced in the early 1980s. In the late 1980s, the double monochromator (Mark III) was developed to reduce the internal instrumental stray light and to improve the signal to noise ratio in large slant column ozone conditions (e.g., to allow high-quality TCO measurements in low sun elevation conditions; to better meet the observation conditions in high-latitude sites). There were other versions of Brewers developed (i.e., Mark IV and V) to extend the measuring wavelengths and to measure other trace gases (Savastiouk, 2006). Today, only the Mark III version of the Brewer is manufactured. More details about Mark II and III measurements and other characteristics are can be found in Zhao et al. (2021).

The Brewer spectrophotometer provides data products that include column ozone (e.g., Kerr, 2002; Kerr et al., 1981), column sulphur dioxide ($SO_2$; e.g., Fioletov et al., 1998; Zerefos et al., 2017), column nitrogen dioxide ($NO_2$, by Mark IV only; e.g., Kerr et al., 1988; Cede et al., 2006; Diémoz et al., 2021), spectral UV radiation (e.g., Bais et al., 1996; Fioletov et al., 2002), aerosol optical depth (AOD) (e.g., Kazadzis et al., 2005; Marenco et al., 2002; Diémoz et al., 2016; López-Solano et al., 2018), and effective ozone layer temperature (Kerr, 2002). However, the main data product provided by Brewer instrument is the TCO via direct-sun (DS) observations using the method described above. In this work, we focus on the Brewer direct-sun TCO data product only, although TCO also can be retrieved using solar zenith-sky radiance, solar global spectral UV irradiance, lunar direct irradiance, and “Focused Sun” method (e.g., Josefsson, 2003; Fioletov et al., 2011; Kerr, 2010).

In this work, Canadian Brewer data was processed by the Brewer Processing Software (BPS) developed by ECCC (Fioletov and Ogyu, 2008). Here we included one world reference triad (BrT-D; Brewer #145, #187, and #191) and one MLO instrument (Brewer #119). European data were processed by European Brewer Network (EUBREWNET) developed by AEMET (Redondas et al., 2018a; Rimmer et al., 2018), including observations from the RBCC-E triad (Brewer #157, #183, and #185), the Swiss triad (Brewer #040, #072, #156), and Brewer #163 (PMOD/WRC instrument). Both BPS and EUBREWNET software demonstrated very similar results and good performance in a recent comparison of available processing software tools for Brewer total ozone retrievals (Siani et al., 2018).

### 2.2 Cimel Sunphotometer

Cimel Sunphotometers are concepted and initiated by the National Aeronautics and Space Administration (NASA) for measuring aerosol properties. The instruments formed a global aerosol monitoring network, known as the Aerosol Robotic Network (AERONET, https://aeronet.gsfc.nasa.gov/ last access: 15 Dec 2022) in the early 1990s (Holben et al., 1998, 2001). The sunphotometer measures AOD in direct sun mode at eight wavelengths, typically 340, 380, 440, 500, 675, 870, 940, and 1020 nm. AOD is a unitless quantity representing the vertically integrated extinction of radiation due to scattering and absorption. The measured AOD values are used for filtering days with high aerosol loading that could affect the Brewer primary calibration accuracy due to scattering and absorption by aerosols. Ideally, observations of AOD at the Brewer instrument's operational wavelength should be used to screen out such days. However, the shortest wavelength of the Cimel Sunphotometer is 340 nm, which also has larger uncertainty than all the other channels (due to a stronger temperature sensitivity). Thus, instead of using the measured Angstrom Exponent to estimate AOD at 320 nm (Brewer instrument's observation wavelength), we used a second-order polynomial fit of measured AOD values (in logarithmic coordinates) from 380, 440, and 500 nm to extrapolate AOD values at 320 nm (personal communication with Thomas F. Eck from NASA Goddard Space Flight Center).

### 2.3 MERRA-2

The second Modern-Era Retrospective analysis for Research and Applications (MERRA-2) is an atmospheric reanalysis from NASA's Global Modeling and Assimilation Office (GMAO). MERRA-2 assimilates partial total column ozone retrievals from the SBUV series from 1980 to 2004 (Labow et al., 2013). From October 2004, MERRA-2 assimilates ozone profiles and total column data from the Microwave Limb Sounder (MLS) and the Ozone Monitoring Instrument (OMI), respectively (Wargan et al., 2017). MERRA-2 column ozone data has been found to be of good quality when compared with satellite and ground-based observations (e.g., Rienecker et al., 2011; Wargan et al., 2017; Zhao et al., 2017, 2019). In this work, the MERRA-2 TCO (0.5° Lat. × 0.625° Lon., version 5.12.4) with 1-hour temporal resolution is used as an input in the ETC simulation model (see Section 3 for more details).

In this study, the MERRA-2 data were compared with real Brewer observations for all 11 instruments from the four sites. In general, the overall percentage difference between Brewers and MERRA-2 is within ±5% on the 1σ level. Small differences in seasonal and diurnal patterns are expected to have only a limited impact on the modelled daily ozone variation pattern or the ETC simulation model (for 2009–2019). The data assimilation source change only included a small shift in bias (1–2%) (Zhao et al., 2021). More details of the comparison results are shown in Appendix A

**2.4 Sites**

Four long-term Brewer calibration and/or operation sites are included in this work: Arosa/Davos, Switzerland; Izaña, Spain; MLO, Hawaii, U.S.A; and Toronto, Canada (see Table 1 for details). MLO and Izaña are the calibration sites for the world reference triads and the European regional triad, respectively. Notably, there are four Brewer triads in operations worldwide, the Swiss Brewer triad (Stübi et al., 2017b), the European regional Brewer reference triad (León-Luis et al., 2018), and two world Brewer reference triads (Fioletov et al., 2005; Zhao et al., 2021). Arosa/Davos, Izaña, and Toronto are the operation sites for the Swiss triad, RBCC-E, and the world references, respectively.

Most of these instruments were operated at their current home site in the last decade, but with some exceptions. One of the Swiss Brewer triad instruments (Brewer #072) was moved from Arosa to Davos in November 2011 (Stübi et al., 2017a). The last instrument (Brewer #040) was moved to Davos in February 2021. RBCC-E started primary calibrations in 2003 and the triad has been in operation at Izaña since 2005, organizing a yearly calibration campaign in Europe (at south Europe El Arenosillo (Huelva) in even years and at Arosa/Davos in odd years). Please note that, as the regional travelling reference, Brewer #185 visited many other field sites over the past few decades (only its record from Izaña is included in this work). The BrT and BrT-D have been in operation in Toronto since 1984 and 2011, respectively (BrT is not included in this work). The BrT-D were temporally relocated to Egbert in 2018–2021. More details of the location and altitude of each site and brief instrument history are shown in Table 1. In short, the relocation of the Swiss triad (Stübi et al., 2017a) and BrT-D would not impact the goal and results of this research (e.g., the new and old sites are within one grid of the MERRA-2 model). Thus, in this work, we will refer to observations and model results from the Swiss triad as being at the Davos site; we will refer observations and model results from BrT-D as being at the Toronto site; i.e., we will not distinguish Arosa and Egbert as separated sites from Davos and Toronto.

**Table 1. Observations sites and instrument history**

| Site | Latitude | Longitude | Altitude (asl) | Instrument (operation periods) | Instrument type |
|---|---|---|---|---|---|
| Arosa[+] | 46.78° N | 9.67° E | 1840 m | #040 (1988 – 2021) | Mark II |
| | | | | #072 (1991 – 2011) | Mark II |
| | | | | #156 (1998 – 2018) | Mark III |
| Davos | 46.80° N | 9.83° E | 2370 m | #163 (2007 –) | Mark III |
| | | | | #040 (2021 – ) | Mark II |
| | | | | #072 (2011 – )[^] | Mark II |
| | | | | #156 (2018 – ) | Mark III |
| Izaña | 28.31° N | 16.50° W | 2370 m | #157 (1997 – ) | Mark III |
| | | | | #183 (2003 – ) | Mark III |
| | | | | #185 (2005 – ) | Mark III |
| MLO | 19.5° N | 155.6° W | 3400 m | #119 (1997 – ) | Mark III |
| Toronto | 43.78° N | 79.47° W | 187 m | #145 (1998 – )* | Mark III |
| | | | | #187 (2007 – )* | Mark III |
| | | | | #191 (2009 – )* | Mark III |
| Egbert* | 44.23° N | 79.78° W | 253 m | | |

[+]Swiss triad (Brewer #040, #072, and #156) was relocated from Arosa to Davos. Arosa site is 13 km south-west of the Davos site. In this work, Brewer data from Arosa and Davos are merged as observations from one site (i.e., Davos). The official name of the observation station is Arosa/Davos.

^From 2011 to 2018, Brewer #072 sometimes moved back to Arosa and again to Davos to perform the intercomparison study (Stübi et al., 2017a). The instrument was at Arosa for four periods (2012 July to August, 2013 February to March, 2014 July to 2015 July, and 2018 July to September).

*World reference triad (Brewer #145, 187, and 191) was temporally relocated from Toronto to Egbert (observations at Egbert from September 6, 2018 to March 8, 2021). Egbert site is 55 km north-west of the Toronto site. In this work, Brewer data from Egbert and Toronto are merged as observations from one site (i.e., Toronto).

## 3 Primary Calibration

### 3.1 Langley plot techniques

Primary calibrations are carried out for Brewer reference instruments independently using the Langley plot technique, which is also called the zero airmass extrapolation technique (Kerr et al., 1985b). With assumptions that TCO values (Ω) are constant through the calibration session (half-day) and aerosol has negligible impact, the instrument response $F_i$ adjusted for instrumental (dead time, dark counts) and some atmospheric (Rayleigh scattering) factors is a linear function of airmass ($\mu_i$):

$$F_i = \mathrm{ETC} + 10\Omega\Delta\alpha\mu_i + e_i \qquad (2)$$

where, $i$ is the observation number, $\Delta\alpha$ is the effective ozone absorption, and 10 is a scaling factor used in the Brewer software, ETC is the extraterrestrial constant (here, ETC = $-10^4 \times F_0$). Equation 2 is a linear regression equation with two unknown parameters, ETC and Ω. That can be solved, for example, by the least squares method, i.e., by minimizing the sum of $e_i^2$. This can be represented by the Langley plot: the linear fit of measurements made under a range of $\mu_i$ (e.g., 1.2 to 3.2; for a short (half-day) period to assure Ω is a constant) can yield a straight line on a $F_i$ vs. $\mu_i$ plot with the intercept as ETC. To find an optimal solution of Eqn. 2, a correlation matrix of $e_i$ is required. In practice, however, it is typically assumed that $e_i$ are uncorrelated and have the same variance. Each Brewer DS measurement comprises five individual observations taken about 41 seconds apart. The standard deviation of these five measurements, std($O_3$), is often used to filter out "bad" measurements (usually, measurements taken under cloudy conditions) and then only the remaining measurements are used to derive ETC. Uncertainties of individual $F_i$ values as well as the dependence of these uncertainties on airmass are not considered as they have a relatively small impact on the final ETC values. Also, the exact time of each of the five individual ozone observations can be used to calculate airmass and the corresponding $F$ values can be used in Eqn. 2 instead of the standard $F$ and airmass from Brewer DS. However, such additional details have a minor impact on the overall ETC uncertainty.

Meanwhile, there is one more factor that is important for consideration. Since the final goal of the ETC estimation is to retrieve the ozone values, it may be more appropriate to minimize the impact of ETC errors on rederived ozone and minimize the sum of ${e_i'}^2$ in a modified equation:

$$\frac{F_i}{\mu_i} = \frac{\mathrm{ETC}}{\mu_i} + 10\Omega\Delta\alpha + e_i' \qquad (3)$$

where the Langley fit will be done with $F/\mu$ versus $1/\mu$, with the slope of the fit as ETC value. In addition, as the Brewer instrument has a constant sampling frequency (about five minutes, if measuring only DS ozone), the Langley fitting plot will have sparse data during high $\mu$ conditions as the sun rises quickly and a dense amount of data while the $\mu$ is lower. Thus, the fitting results of Eqn. 2 might be over-weighted by those sparse observations near the early morning or late afternoon periods, whereas Eqn. 3 does not have such a problem. Preference of Eqn. 3 over Eqn. 2 is also to consider the important assumption that ozone remains constant during the observation period. However, if ozone should fluctuate, this is likely to be the main source of errors ($e_i$ and $e'_i$ in Eqns. 2 and 3) in the Langley fits. In Eqn. 2, fluctuations in ozone are not uniform over the fit because they are enhanced by the factor $\mu$. However, in Eqn. 3, errors caused by fluctuations in ozone are uniform over the

entire fit. A particular form of the regression model (Eqn. 2 or Eqn. 3, or similar) may produce somewhat different values for the ETC; however, the differences are typically small as discussed below (e.g., Kiedron and Michalsky, 2016).

The Brewer instrument is able to utilize up to five neutral density (ND) filters to attenuate the incoming light to be within the dynamic range of its photomultiplier tube (PMT). Although the filters are selected to have uniform responses (i.e., they attenuate all wavelengths nearly equal), the ETCs for different ND filters (i.e., fitting results of Langley plots made with observations via different ND filters) still can have offsets. Thus, observations via different ND filters can be grouped and fitted with the "shared intercept" method:

$$\frac{F_i}{\mu_i} = \sum I_{ij} \frac{\mathrm{ETC_j}}{\mu_i} + 10\Omega\Delta\alpha + e_i' \qquad (4)$$

where, $I_{ij}$ is the index matrix that selects observations made with ND filter number $j$ (i.e., if $F_i$ is observation with ND $j$, $I_{ij}$ = 1, otherwise, $I_{ij}$ = 0). By using this adapted method, ETC values for individual ND filters will be fitted, with a shared intercept value (10ΩΔα; as a constraint that observations via different ND filters should report the same constant total ozone value for the observation period).

Note that the fitting method and the data selection criteria are somewhat different between the AEMET and ECCC teams. For example, the AEMET procedure uses $\mu$ range from 1.75–3.75 (which is to choose an airmass range that covers all year around), while the ECCC procedure uses 1.2–3.2. In practice, the AEMET procedure uses both Eqns. (2) and (3) to fit ETC values, while the ECCC procedure uses Eqn. (4). Note that, for the AEMET procedure, the ND correction is applied via a different approach (Redondas et al., 2018a). The detailed summary of the difference between these procedures is a subject of a separate upcoming study, and is not included here. In this work, we only focus on the impact of selections of fitting equations.

Figure 1a shows the individual ETC values fitted using Brewer #119 observations for a seven-month period (2010 April to October) at MLO. Here Eqns. 3 and 4 were used in the analysis (represented by different colours for each equation in the figure), but with other fitting factors set to common values (e.g., using a common $\mu$ range; 1.2–3.2). For Eqn. 4, the individual ETC values are results from ND filter no. 4 (which is the most frequently used ND filter in MLO, given its local sunlight conditions). The mean values of ETCs and the standard deviations are shown in the legend. Figure 1b shows the fitting results of the intercept term in Eqns. 3 and 4. Figure 1c shows the Langley fitting residual, with their mean residual values reported in the legend.

For a Brewer instrument, the goal of such calibration is to derive its unique ETC value (that can be used in TCO calculation), with uncertainty within ±5 R6 units (Zhao et al., 2021). Here R6 is a measurement-derived double ratio in the actual Brewer processing algorithm, corresponding to the measured slant column ozone (e.g., Savastiouk, 2006; Zhao et al., 2021). In general, Fig. 1 shows that the selection of fitting equations will not impact the agreement of the final averaged ETC, as long as an adequate number of individual ETCs are included. In addition, when short-term ozone variability is high (i.e., indicated by large day-to-day variation of the $10\Delta\alpha\Omega$ term shown in Fig. 2b), the individual ETCs will have slightly larger variability (e.g., see results from late April to early June). For example, Fig. 1 shows that even for a selected fitting method described above, its individual ETCs have up to 15 R6 unit standard deviations (which correspond to 0.68–0.80% of TCO changes, in typical conditions; $\Omega$ = 280 DU, $\Delta\alpha$ = 0.34, and $\mu$ = 2). This large day-to-day difference is mainly due to short-term ozone variability. Note that, in all three Langley plot techniques (Eqns. 2–4), the assumption was the TCO ($\Omega$) for the measurement period is a constant value. Typically, if the TCO changes' amplitude is ≤1.5 DU for a calibration session, the fitted individual ETC value can meet the calibration goal (i.e., uncertainty within ±5 R6 unit). However, without knowledge of true atmospheric TCO variation conditions, it is difficult to accurately determine if such desired calibration conditions were met.

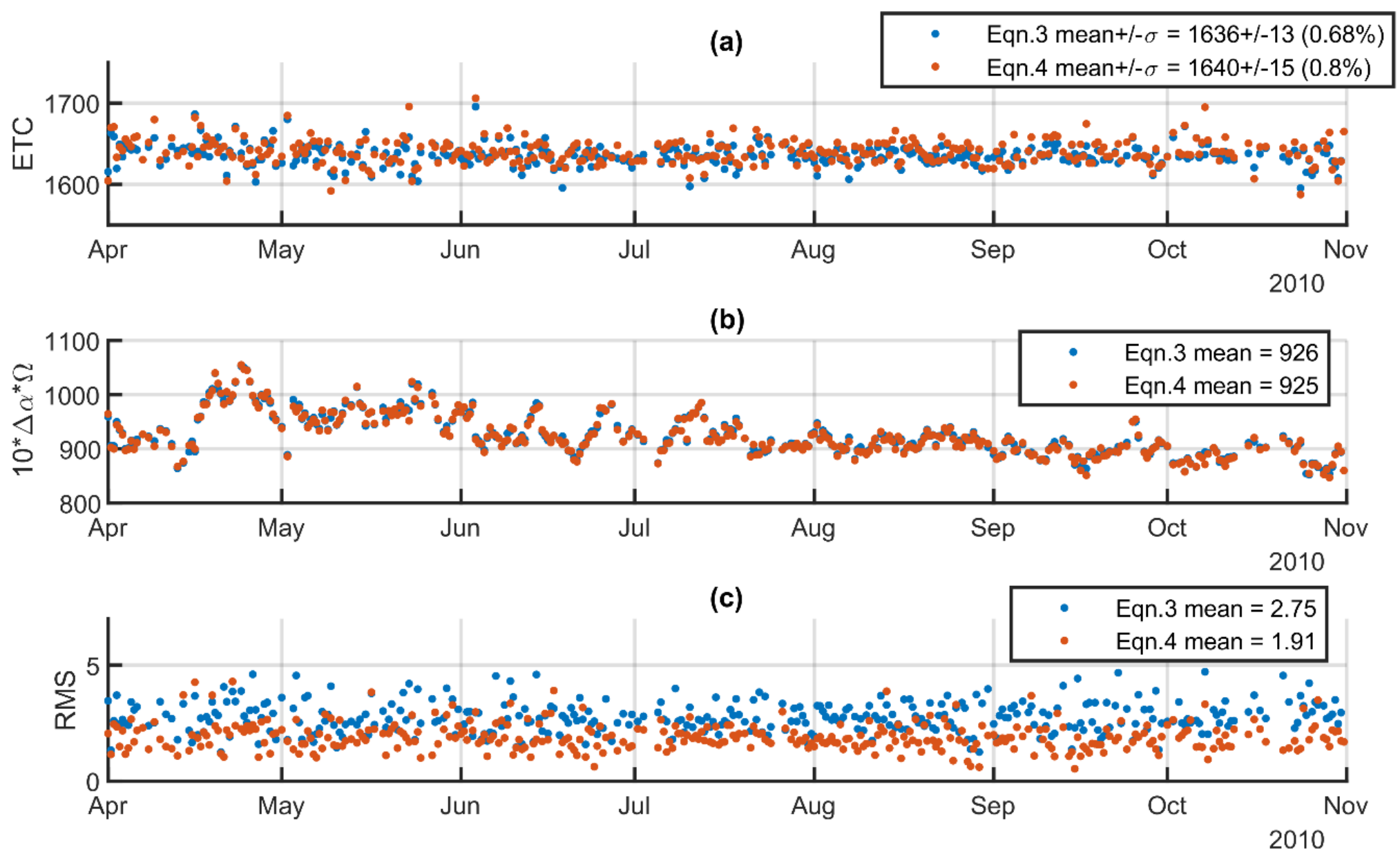

**Figure 1. Brewer #119 individual ETC fitting results from April to October 2010 at MLO, (a) individual ETC values, (b) the intercept term of Langley fits (for Eqns. 3 and 4), (c) Langley fitting residual RMS. Each dot represents the results of one half-day estimate. The percentage value in panel (a) legend is the 1σ value corresponded % of TCO changes, in typical conditions ($\Omega$ = 280 DU, $\Delta\alpha$ = 0.34, and $\mu$ = 2).**

### 3.2 Primary calibration method

As illustrated in the previous section, in reality, TCO does not remain constant during the calibration session (half-day, i.e., short-term ozone variability) which will result in an uncertainty in a single fitted ETC value using any equations described above. Thus, to lower the ETC uncertainty, Langley calibrations (single fitted ETC values) are, therefore, averaged over a certain period (e.g., 5–10 days for the MLO site) to reduce the effects of short-term ozone variability (Kerr, 2010). We will refer to this final averaged ETC value as the ETC product for $O_3$ ($ETCO_3$) and fitting results of individual calibration sessions (i.e., one morning or one afternoon period) as individual ETC measurements. This $ETCO_3$ will be used to calculate observed ozone columns via $\Omega = \frac{R6 - ETCO_3}{10\Delta\alpha\mu}$. The instrument receiving its $ETCO_3$ via this method is considered to be calibrated by the PCM. In contrast, an instrument receiving its $ETCO_3$ by comparing its observations with another calibrated Brewer is considered to be calibrated by the CTM (i.e., by using the same ozone column equation, but replacing the Ω with observations from another calibrated Brewer, one can derive the $ETCO_3$). In this work, we will focus on the PCM only.

As the calibration site for world reference Brewers, MLO's short-term ozone variability conditions have been well studied and known (Gröbner and Kerr, 2001; Kerr, 2010). Kerr (2010) points out TCO does not remain constant at MLO, having a typical variation of about ±2 DU during a day. This variation would contribute about 1.5% error in total ozone if using an individual ETC (without averaging) to calculate the TCO. Gröbner and Kerr (2001) concluded that, for Brewer and Dobson instruments at MLO, the PCM can be carried out at any time of the year. For other observation sites, this short-term ozone variability can be very different from that for MLO and would introduce different amounts of uncertainties to individual ETCs and $ETCO_3$. We do not have similar quantified numbers for other sites; however, it can be noted that, typically, the low-latitude sites will have less short-term ozone variability. In addition to latitude-dependency, the short-term column ozone variability also has a seasonal dependency, showing lower variability in the summertime. Thus, even for a given site, if the PCM is carried out in different seasons, the derived individual ETCs and $ETCO_3$ would also have different uncertainties. Therefore, a quantification of the short-term ozone variability's impact on the Brewer's primary calibration work is necessary.

For consistency, moving forward, all PCM calculations are done using Eqn. 2 and $\mu$ range from 1.2–3.2. The individual fitting results are then filtered with the same criteria (≥ 10 good quality ozone data points per each fit and the least square fitting RMS ≤ 3). The long-term ETC fitting results from all 11 instruments are shown in Fig. 2. Blue dots are the fitted individual ETCs, while the red lines represent the yearly normal fits of corresponding individual ETCs (with error bars as one standard deviation of the individual ETCs of the particular year). The documented ETC values from the instrument calibration file (ICF) are shown as green lines with their validation period indicated by vertical black dash lines. The first column of the figure (left to right) shows results from instruments at Arosa/Davos; the second column, RBCC-E instruments; the third column, MLO instrument; and the last column, world reference instruments.

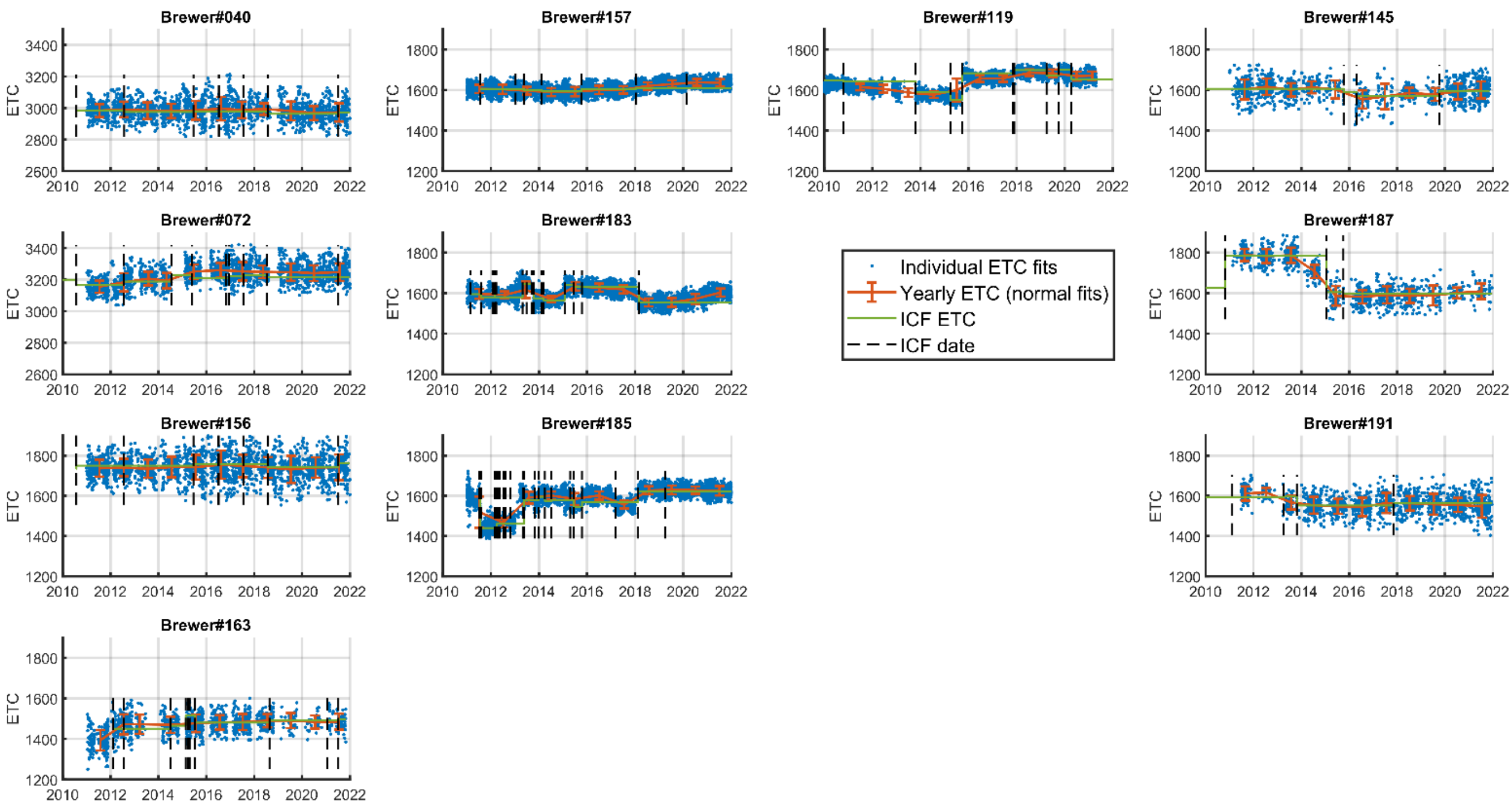

**Figure 2. Long-term Langley fits of Brewers from Arosa/Davos (Brewers #040, #072, #156), Davos (Brewer #163), Izaña (#157, #183, #185), MLO (#119), and Toronto (#145, #187, #191). Blue dots are fitted ETCs for individual half-days, red lines are yearly mean values of these individual ETCs (error bars are 1σ values), and green lines are ETC values obtained from instrument calibration files (ICF; validation periods are indicated as vertical black dash lines).**

Even before any further detailed analysis, Fig. 2 already shows the quality of individual ETCs from different sites via their spread patterns. For low-latitude sites (Izaña and MLO), the individual ETCs have less variability and are more closely distributed around the yearly means and ETC values from ICF. For mid-latitude sites (Arosa/Davos and Toronto), the individual ETCs are more scattered and even show seasonal patterns (i.e., higher scattering in cold seasons).

## 3.3 ETC simulation model

The scattered individual ETCs as shown in Fig. 2 came from several aspects, including short-term ozone stability, instrument random uncertainty, meteorological impact (e.g., missing data due to cloudy or high aerosol conditions), and also instrumental function changes in individual Brewers (e.g., changes in optics or other electronic parts). For example, Brewer #187 has a clear change of $ETCO_3$ from 1800 in 2013 to about 1600 in 2015. During this period, it was discovered that the PMT quality was poor and had extremely high dark counts. Adjustments were made to try to compensate for the deterioration and reduce the dark count of the PMT. In September 2014, the PMT was replaced. It was at this point the instrument also had its electronics

converted (from single-board electronics to multi-board) which also affected its $ETCO_3$. More detailed examples of instrumental changes of RBCC-E instruments are provided in the supplement of León-Luis et al., (2018). To avoid such instrument-related issues and meteorological factors, MERRA-2 reanalysis ozone data are used in an ETC simulation model developed to isolate the short-term ozone variability impacts and also to avoid data gaps. We developed such a model to provide a uniform workbench to assess any observation locations that may be considered for Brewer's primary calibration work and provide guidance for Brewer’s long-term stability monitoring (addressed in Section 5).

The ETC simulation model uses MERRA-2 TCO as input. MERRA-2 data were found to have good consistency with Brewer observations (e.g., Zhao et al., 2021). The data used comprises 20 years of MERRA-2 reanalysis hourly data (1999–2019) for Davos, Izaña, MLO, and Toronto (detailed comparison and verification can be found in Appendix A). For each day (local standard time from 6 a.m. to 6 p.m.), MERRA-2 TCO data (Ω) are fitted by a simple 2nd-order polynomial function as it was previously done in several Brewer triad studies (Fioletov et al., 2005; Stübi et al., 2017b; León-Luis et al., 2018; Zhao et al., 2021),

$$\Omega = a + b(t - t_0) + c(t - t_0)^2 \quad (5)$$

where, $t$ is the corresponding time of MERRA-2 ozone, $t_0$ is the time of local solar noon. The fitting results for each site are summarized in Fig. 3 as histograms of the fitting parameters. The analysis shows that both MLO and Izaña have low year-round ozone variations ($a$ term; see Fig. 3a) with comparable mean and standard deviation values (268 ± 17 DU and 291 ± 20 DU, respectively). Arosa/Davos and Toronto have much larger mean and standard deviation values (316 ± 38 DU and 328 ± 42 DU, respectively) when compared to MLO and Izaña due to their higher latitude. The first-order and second-order polynomial terms ($b$ and $c$ terms; see Fig. 3b and c) show the linear and quadratic changes of the ozone field, which represent two independent (orthogonal) TCO variation patterns. For example, on the 1-sigma level, MLO has only 0.48 DU/hr linear ozone field changes, which is less than one-third of the linear variation in Toronto. Simple TCO variation amplitudes (i.e., maximum value subtract minimum value for that given period) for whole daytime, a.m., and p.m. sessions are reported in Fig. 3d to f, with the probability of variation amplitudes less than 1.5 DU (suitable conditions for Brewer Langley calibration work) shown in the legends. For example, the results (Fig. 3e) show that Brewers at MLO and Izaña have a better chance (e.g., 55% and 45% for morning sessions, respectively) to have a good stable short-term ozone field to produce high-quality individual ETCs via Langley plot techniques than the ones at Davos (23%) or Toronto (20%).

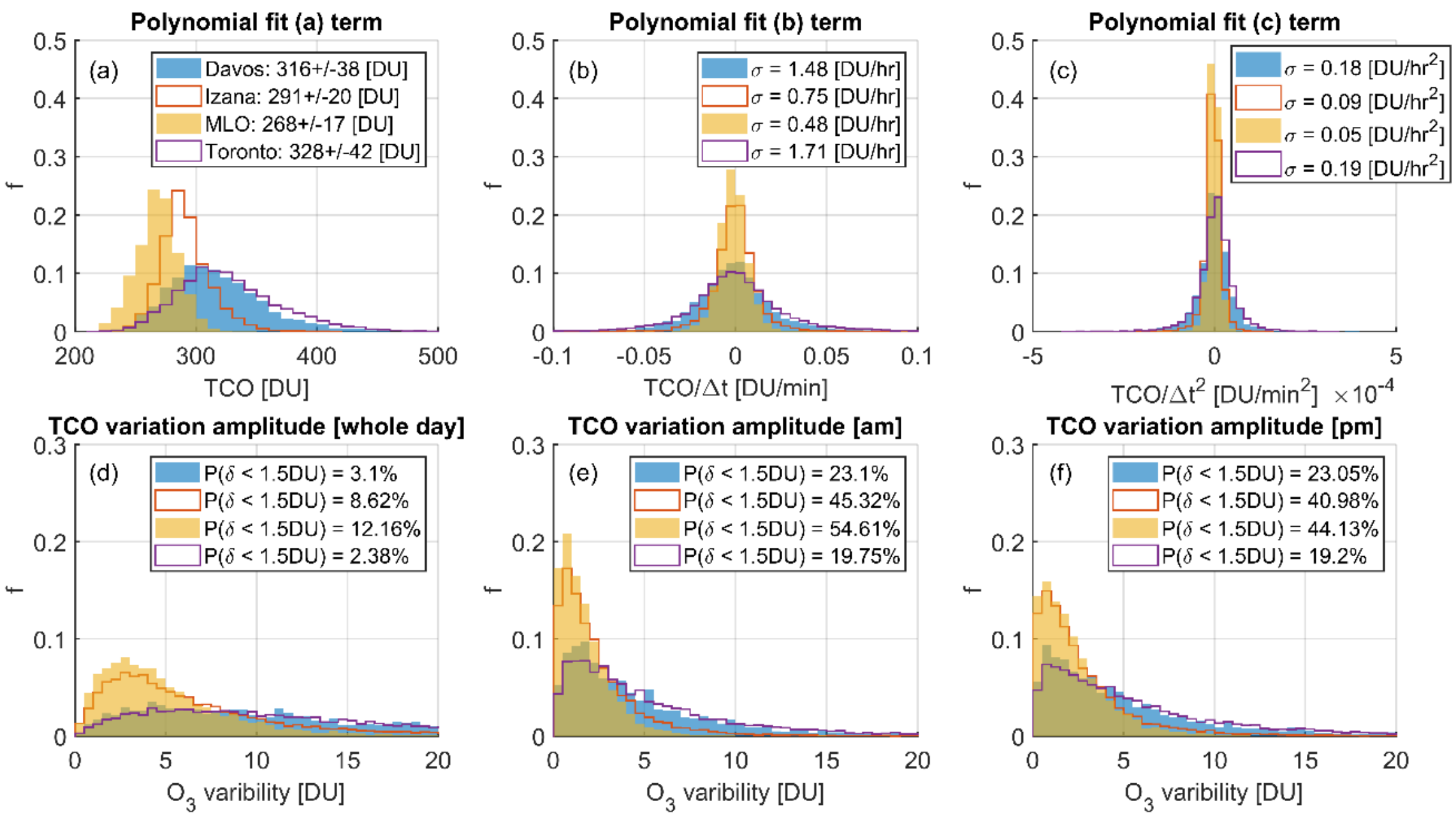

**Figure 3. Short-term ozone variability for Davos, Izaña, MLO, and Toronto sites, as derived from MERRA-2 data. The distribution of the three simple 2nd-order polynomial fitting parameters is shown in panels (a) to (c). The distribution of TCO variation amplitudes are shown in panels (d) to (f), with the probability of TCO variation amplitude less than 1.5 DU indicated in the legends.**

In addition to quantifying the short-term ozone variability at different sites, the benefit of fitting the ozone field using Eqn. 5 is that fitted ozone values can be used to assess and quantify short-term ozone variability's impact on the Brewer individual ETC calculation. Using Eqn. 5, if we assume the fitted ozone values are the true ozone in the atmosphere (Ω) and select a real ETC ($ETC_{real}$) and absorption ($\Delta\alpha$) for the simulation (as a dummy Brewer instrument; see Eqn. 6), the instrument response (simulated $F$ value; $F_{sim}$) can be derived and then used in the Langley fitting (see Eqn. 7, to derive a simulated individual ETC; $ETC_{sim}$).

$$F_{sim} = \mathrm{ETC}_{\mathrm{real}} + (a + b(t - t_0) + c(t - t_0)^2)10\Delta\alpha\mu + e_i \qquad (6)$$

$$\frac{F_{sim}}{\mu} = \frac{\mathrm{ETC}_{\mathrm{sim}}}{\mu} + 10\Omega\Delta\alpha + e_i' \qquad (7)$$

This $ETC_{sim}$ can be directly compared with $ETC_{real}$, and the impact of short-term ozone variability on Brewer's primary calibration then can be quantified. In addition, by including or removing the $a$, $b$, and $c$ terms in Eqn. 6, the impact of linear and quadratic ozone variations can be independently assessed. For example, if the $c$ term in Eqn. 6 is set to 0, one can assess

the $ETC_{sim}$ that only has linear ozone variability contribution. It was found that the linear ozone variability component contributed more to the individual ETC's variability than the quadratic term (depending on the site, linear term induced ETC standard deviations are 1–5 R6 units higher than the ones that are induced by the quadratic term).

To validate this ETC simulation model, $ETC_{real}$ and $\Delta\alpha$ were selected to be 1640 and 0.34, respectively (close to Brewer #119's values in 2010, see Fig. 1a). The simulation results for the same period (April to October 2010) show good agreement with Brewer #119's individual ETC fitting results using real observations (Fig. 4 versus Fig. 1a). In Fig. 4, the simulated individual ETCs for morning and afternoon sessions are reported separately. The results show the simulated individual ETC values also have standard deviations of 9 to 12 R6 units (see Fig. 4a) simply due to this short-term ozone variability. Within these calculations, $F_{sim}$ did not account for instrumental random noise, thus the simulated individual ETC values have smaller standard deviations compared to the real observations (standard deviations of 13 to 15 R6 units; see Fig. 1a). If we include random noise based on the reported precision of the Brewer reference triad (Fioletov et al., 2005; Zhao et al., 2016, 2021), the simulated ETCs variation will be more comparable to the real observations (11 to 15 R6 units; see Fig. 4b). The calculations then had the random noise added directly to the simulated instrument responses (i.e., $F'_{sim} = F_{sim} + e_{random}$) for each one of the calibration periods independently (a whole morning or afternoon session), where $e_{random}$ has a normal distribution with zero mean and standard deviation of 11 R6 units (corresponding to 0.5% precision of final TCO data in typical conditions). Note that here we used the same μ range (1.2–3.2) and other filters as for the real Brewer instrument.

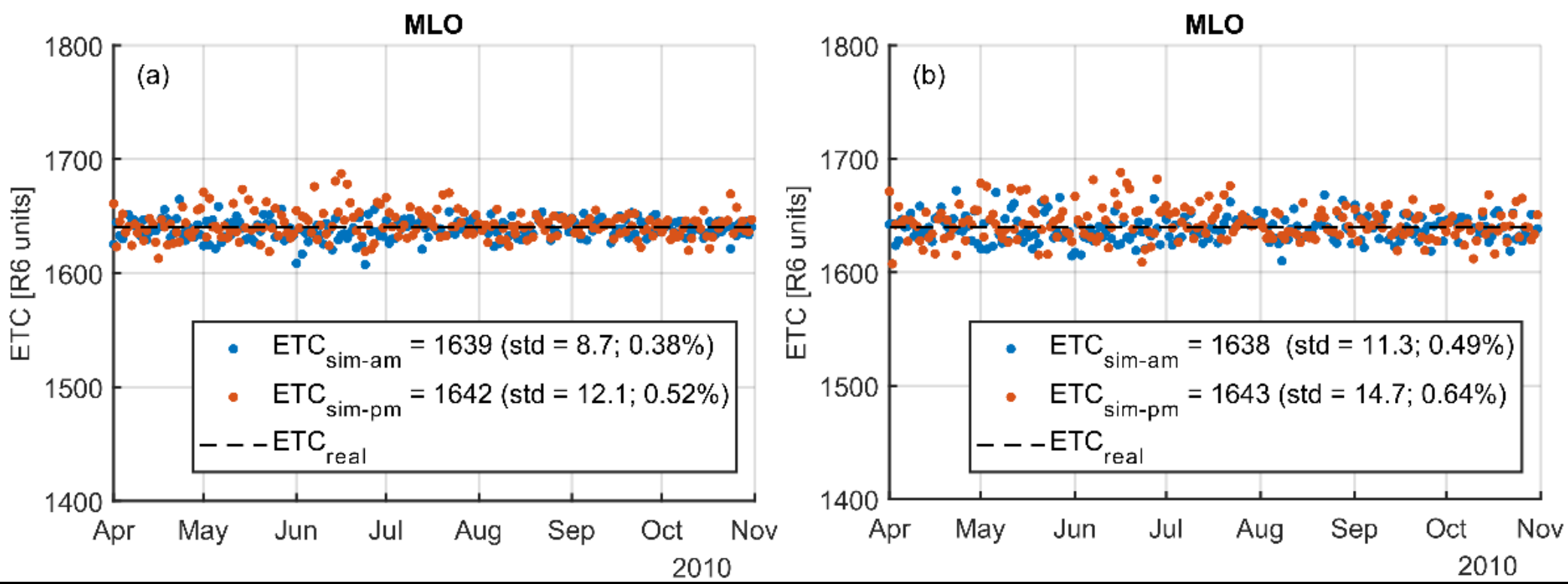

**Figure 4. ETC responses to short-term ozone variations for MLO site (from April to October) simulated from MERRA-2 data. Blue and red dots are simulated individual ETCs (for morning and afternoon sessions, respectively) that would be obtained by a dummy Brewer (with known $ETC_{real}$ as 1640, indicated as the black dash line), with TCO conditions provided by MERRA-2. Panel (a) is the model simulation without instrumental random noise, panel (b) is the simulation with random noise. The mean and 1σ values are shown in the legend. The percentage values are the 1σ value corresponded % of TCO changes, in typical conditions (Ω = 280 DU, $\Delta\alpha$ = 0.34, and μ = 2).**

We applied the same analysis to the Davos, Izaña, MLO, and Toronto sites, using 20 years of MERRA-2 data (1999–2019). Figures 5a to 5d show the monthly mean of modelled individual ETCs (blue and red error bars, represent a.m. and p.m. simulation results), which have a clear seasonal variation in terms of the individual ETC's standard deviations (as shown by the size of error bars). For example, due to stronger short-term ozone variability in Toronto in wintertime, one can have reasonably good monthly mean ETCs as 1640 and 1637 (for morning and afternoon sessions, respectively; see blue and red lines) by averaging all ETC values made in December, but their standard deviations are very high (145 and 146 R6 units). In contrast, in August, the monthly mean ETCs are 1640 ± 28 and 1640 ± 30 (mean ± 1$\sigma$, for morning and afternoon sessions, respectively). There are two implications from these results. First, in practice, it will be very challenging to calculate a high-quality $ETCO_3$ from the primary calibration method at mid- and high-latitude sites (i.e., short-term ozone conditions are more likely to be unstable; the number of days that meet the conditions is small, especially in winter). Second, even if one can derive an $ETCO_3$ by using such scattered individual ETC values, its uncertainty will be much higher than what can be achieved in a low-latitude site (with all other conditions being the same). The results from Davos are similar to Toronto, mainly due to similar short-term ozone variability which is largely a characteristic of the site's latitude. In comparison, MLO and Izaña have much better short-term ozone stability, especially during the summer months. Since MLO is at a low latitude, it is an ideal site for Brewer primary calibrations with an exceptionally stable year round ozone field.

In addition to modelled monthly ETCs, the observation-derived individual ETCs from Brewer instruments (i.e., results from Fig. 2) are also used to calculate monthly ETCs' standard deviations (i.e., individual ETCs subtracted from their respected monthly mean value, and then their standard deviations calculated via the group of months). The calculated observation-derived standard deviations are then added back to $ETC_{real}$ and plotted as green lines in Fig. 5. Note that the green lines in Fig. 5 are results from only one of the Brewers at a given site. For example, the results for the Davos site are coming from Brewer #156. The results of the other Brewers at each site are provided in Appendix B.

As described in Section 3.2, the $ETCO_3$ product of the primary calibration work is a single derived ETC value that will be used in the Brewer TCO calculation. This $ETCO_3$ is the result of averaging several individual ETCs that are obtained in exceptional conditions (e.g., stable short-term ozone, low AOD, good weather). Thus, the uncertainty of the $ETCO_3$ can be calculated as the uncertainty of the mean, for which we have a clear goal (i.e., ±5 R6 unit) (Zhao et al., 2021). Based on this information, the number of sessions (i.e., half-day; $N$) that is needed to meet the calibration goal is derived (see Figs. 5e to 5h; $N \geq \sigma^2/25$, where $\sigma$ is the standard deviation of all individual ETCs used in the average). For example, at MLO, without considering other aspects of the calibration work, one would need about five good calibration sessions to derive a high-quality $ETCO_3$ in October. In contrast, this number will increase to 10 to 14 sessions if the calibrations were performed in December. The results indicate that (1) summertime with lower short-term ozone variability conditions is ideal for Brewer's primary calibration work, (2) low-latitude sites (MLO and Izaña) have better conditions over the mid-latitude sites (Davos and Toronto). Note that, with a

well-maintained Brewer reference instrument, one good primary calibration would be good for its operations for several years if no instrumental issues occurred (e.g., changing of optics). In practice, a two-year calibration period is recommended by GAW.

In summary, Fig. 5e to h show that the ETC simulation model successfully reproduced the general features seen from observed data, in terms of the number of sessions needed at different sites to derive high-quality $ETCO_3$. The minor difference between modelling results and real-observation-derived results is the product of multiple influences, such as data gaps and other meteorological factors that will be discussed in the next section. In short, the blue and red lines (model results) in Fig. 5 represent the sole impact from short-term ozone stability, while the green line (observations) represents a combination of many other factors (that lead to higher standard deviations and resulting number of sessions needed).

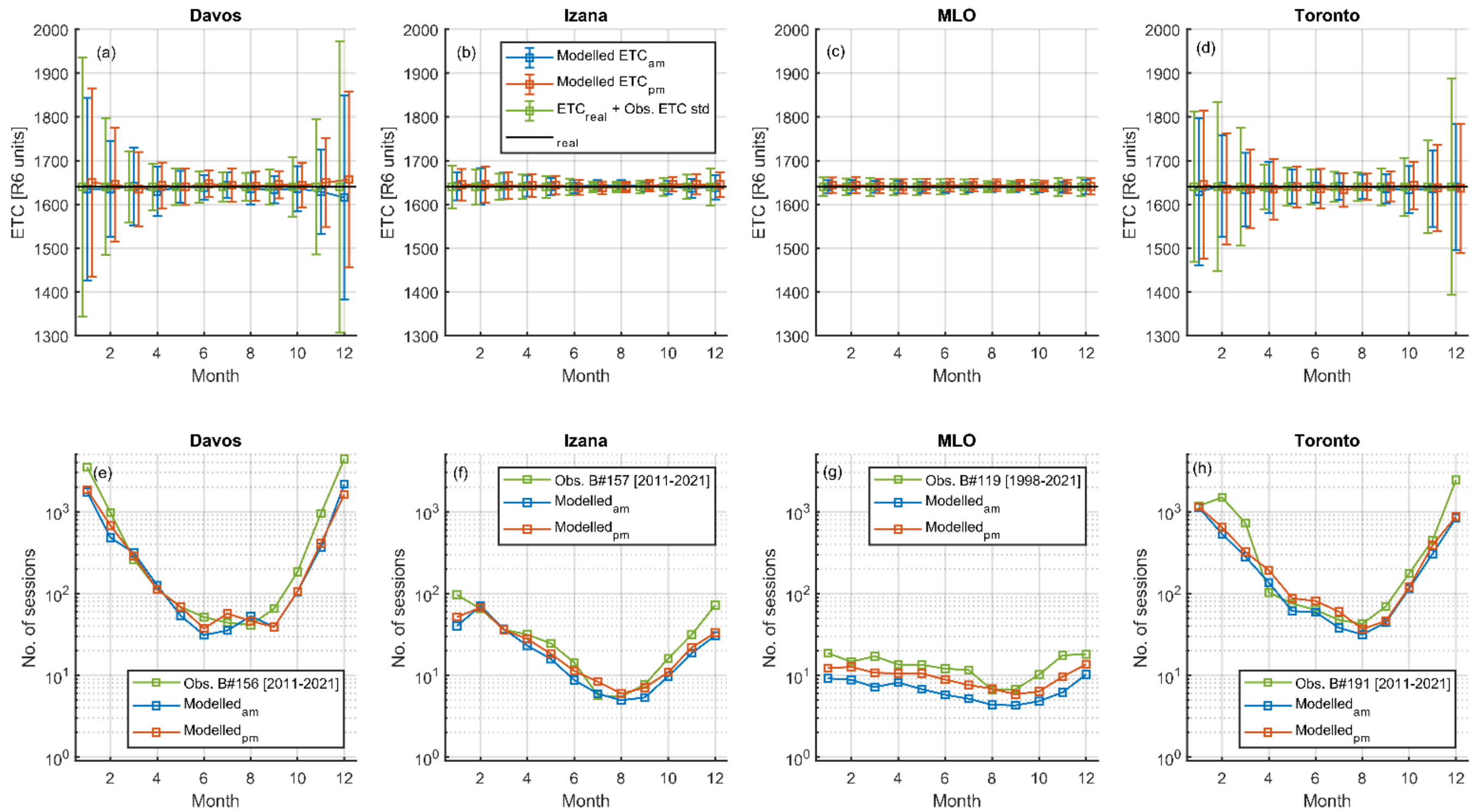

**Figure 5. Monthly mean ETC values (top row) and the number of sessions needed to achieve desired precision of the final ETC product (bottom row) for four sites. On the top row, the imaginary Brewer's true ETC ($ETC_{true}$) is shown as black lines; blue and red lines show the modelled monthly mean ETC with its standard deviations (std); green lines show the $ETC_{real}$ with observation-derived ETC std.**

Another aspect of the results that should be highlighted is these modelled individual ETCs are based on MERRA-2 reanalysis data, which assimilated satellite observations. Note that satellite data have limited sensitivity to low altitude (surface and lower troposphere) variations of ozone, which also contributed to about 10% of TCO. Such variations could occur as a result of air pollution and are even impacted by climate changes (e.g., near populated sites such as Toronto). Such influences could cause random variations or perhaps systematic variations that depend on solar zenith angle (SZA) and other factors.

## 4 Other Site-specified calibration conditions

In addition to short-term ozone stability, several other site-specified conditions could affect the primary calibration accuracy and precision in the process determined. To quantify their impacts, other factors, including AOD and cloudy conditions, have also been evaluated for the four sites using AERONET data and Brewer DS observations. For example, on a yearly average, MLO has a 77% and 99% chance to have good weather and AOD conditions for primary calibration work, respectively (see Fig. 6). However, the cloudy and AOD conditions also might have seasonal patterns. Thus, understanding such cloudy and AOD conditions of a given site can help us achieve the calibration goal.

### 4.1 Aerosol and cloudy conditions

As discussed in Section 3.2, there are several other site-dependent factors that could affect Brewer's primary calibration work. Although Brewer DS ozone retrievals have low sensitivity to atmospheric aerosols (i.e., for AOD 320 nm = 1, induced TCO uncertainty is typically within 1 DU), very high aerosol loading conditions may affect the calibrations and should be avoided. For example, to reduce the impact of the dust storm on ozone measurements, the AEMET team removed data collected when AOD was high (i.e., half-day mean AOD via AERONET observations at 340 nm $\leq 0.5$).

Figures 6a to 6d show the probability of good AOD conditions (AOD 320 nm < 1 as solid lines; AOD 320 nm < 0.5 as dashed lines, for reference) at the four sites to perform primary Brewer calibrations. MLO has very good (AOD < 0.5) conditions almost year-round, while the Izaña site has slightly fewer good sessions for calibration work in July and August due to the Saharan dust (Rodríguez et al., 2015). The Toronto site is in an area where AOD conditions are worse in warm seasons, but has > 95% chance to have AOD conditions < 1 for calibration work.

As mentioned, std($O_3$) can be used to screen out cloudy conditions. For a given half-day, if there are more than 10 successful DS ozone observations with std($O_3$) $\leq 2.5$, the period is considered to have acceptable weather conditions (for Brewer calibration). Figure 6e to 6h show the probability of such good weather conditions calculated as a fraction of half-days with more than 10 successful DS to the total number of half-days. At MLO, the west side of the island is mainly sunny, and the east side of the island is often cloudy due to the prevailing easterly wind. The MLO site has more cloudy conditions in the afternoon

(see Fig. 6g) since daytime heating promotes warm moist air from the ocean to progress up the mountain and form into clouds during the afternoon. In contrast, Izaña has prevailing north-easterly trade-winds that bring more clear-sky conditions. Thus, Izaña has more good sunny conditions for calibration work (see Fig. 6f). Davos and Toronto have similar weather conditions in warm seasons, while Toronto has more cloudy days in winter. For these two sites, besides poor weather conditions, the limited SZA range in the winter months makes them unsuitable for any Langley calibration work during that time.

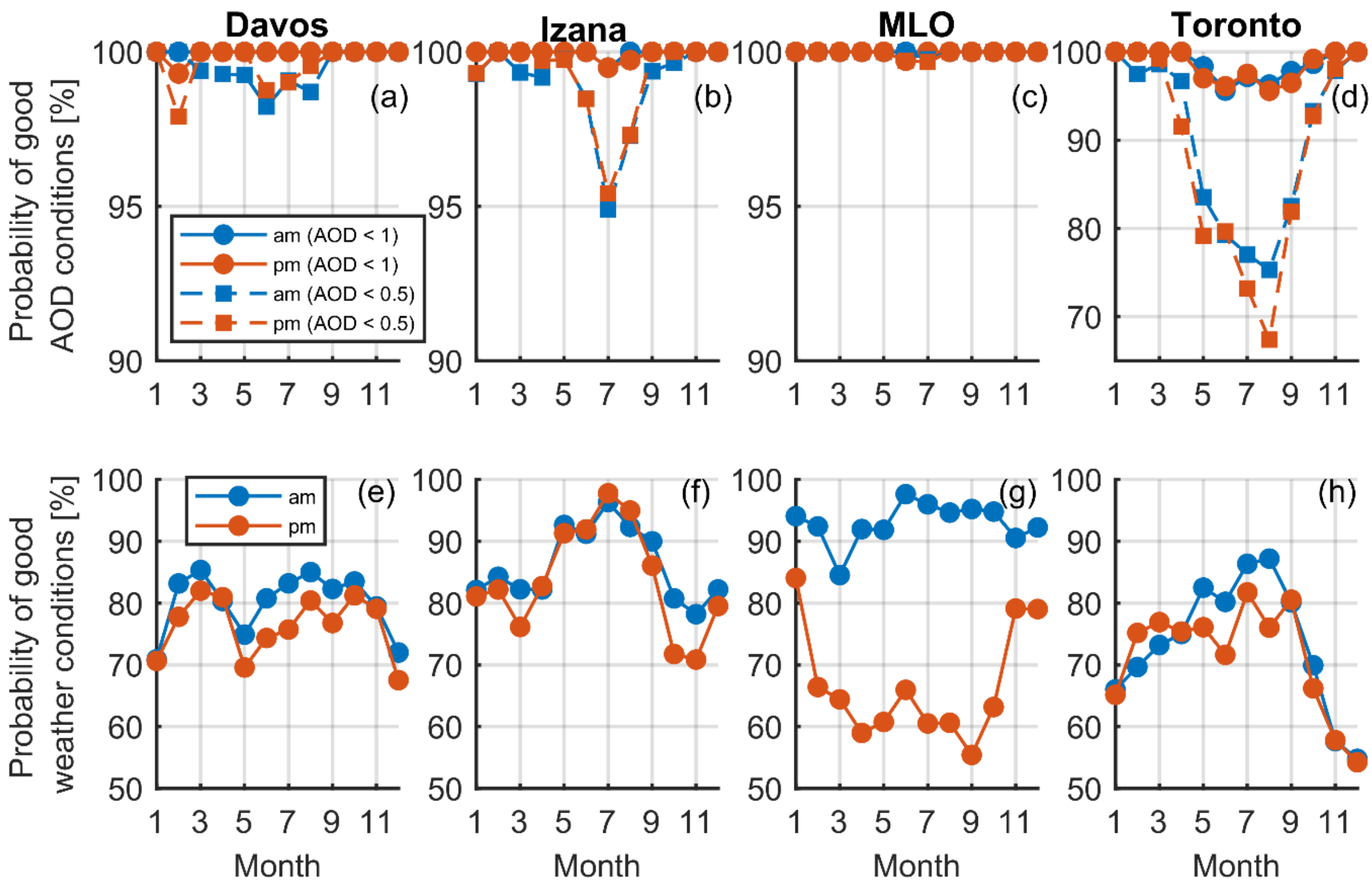

**Figure 6. Probability of good AOD and weather conditions for Brewer's primary calibration at four sites in 2005–2020. AOD conditions were estimated via sunphotometer observations, while weather conditions were estimated via Brewer direct-sun ozone observations. Note panel (d) used a different y-axis scale than panels (a) to (c).**

### 4.2 Total number of days needed for primary calibration work

Accounting for these site-dependent factors (site characteristics; short-term ozone stability, AOD, and cloudy conditions) that can affect Brewer's primary calibration work, Fig. 7 shows the number of days that is required for MLO and other sites to deliver high-quality $ETCO_3$ estimated from model simulation and real observations. Here, the number of sessions is converted to the number of days (i.e., there are two sessions for each day; morning and afternoon sessions). For model results, the number of days needed is estimated as:

$$N_{day}(m) \geq \frac{1}{p_{low-AOD}(m)} \times \frac{1}{p_{non-cloudy}(m)} \times \frac{\sigma^2_{model}(m)}{50} \quad (8)$$

where $p_{\text{low-AOD}}$ is the probability of good AOD conditions, $p_{\text{non-cloudy}}$ is the probability of good non-cloudy weather conditions, $\sigma_{\text{model}}$ is the standard deviation of all modelled individual ETCs for a month ($m$). For the observation results, the number of days needed is estimated as $N_{day}(m) \geq \sigma_{\text{obs}}(m)^2/50$, where $\sigma_{\text{obs}}$ is the standard deviation of all measurement-derived individual ETCs for a month ($m$). Note here, for the real observation derived results, there is no need to include the probabilities of good AOD and non-cloudy conditions since the default data quality control filters for Langley fits already have these included (e.g., for real observation data, we only do Langley fits if there are ≥ 10 observations for a session).

For all four sites, the simulation lines (blue) have a u-shape with an increased number of days in wintertime. The observation lines (red) have a similar pattern as the simulation lines, but are noisier due to instrumental and other factors (e.g., observation gaps). Figure 7 shows that, typically, the summer months have the best conditions for primary calibration work, when the TCO field is stable. The calibration conditions at low- and mid-latitudinal sites are very different. For the Davos and Toronto sites, the lines have sharper curvature in cold seasons, which indicates the challenges of performing primary calibration work at mid-latitude sites. For these two sites, the variable TCO field, cloudy days, as well as the limited SZA in winter months make them unsuitable for any primary calibration work. The number of days required is greater than the number of days available for most months of the year. More importantly, this work confirms and quantifies the conditions at both world and European regional reference Brewers' calibration sites (i.e., MLO and Izaña), showing they have premium conditions to perform Brewer primary calibration work.

Meanwhile, due to the different characteristics of each site, different optimized strategies must be selected by the AEMET and ECCC Brewer teams in practice. For example, it was found the MLO site has more cloudy conditions during afternoon sessions and thus, more observations must be made to ensure a good balance of morning and afternoon sessions.

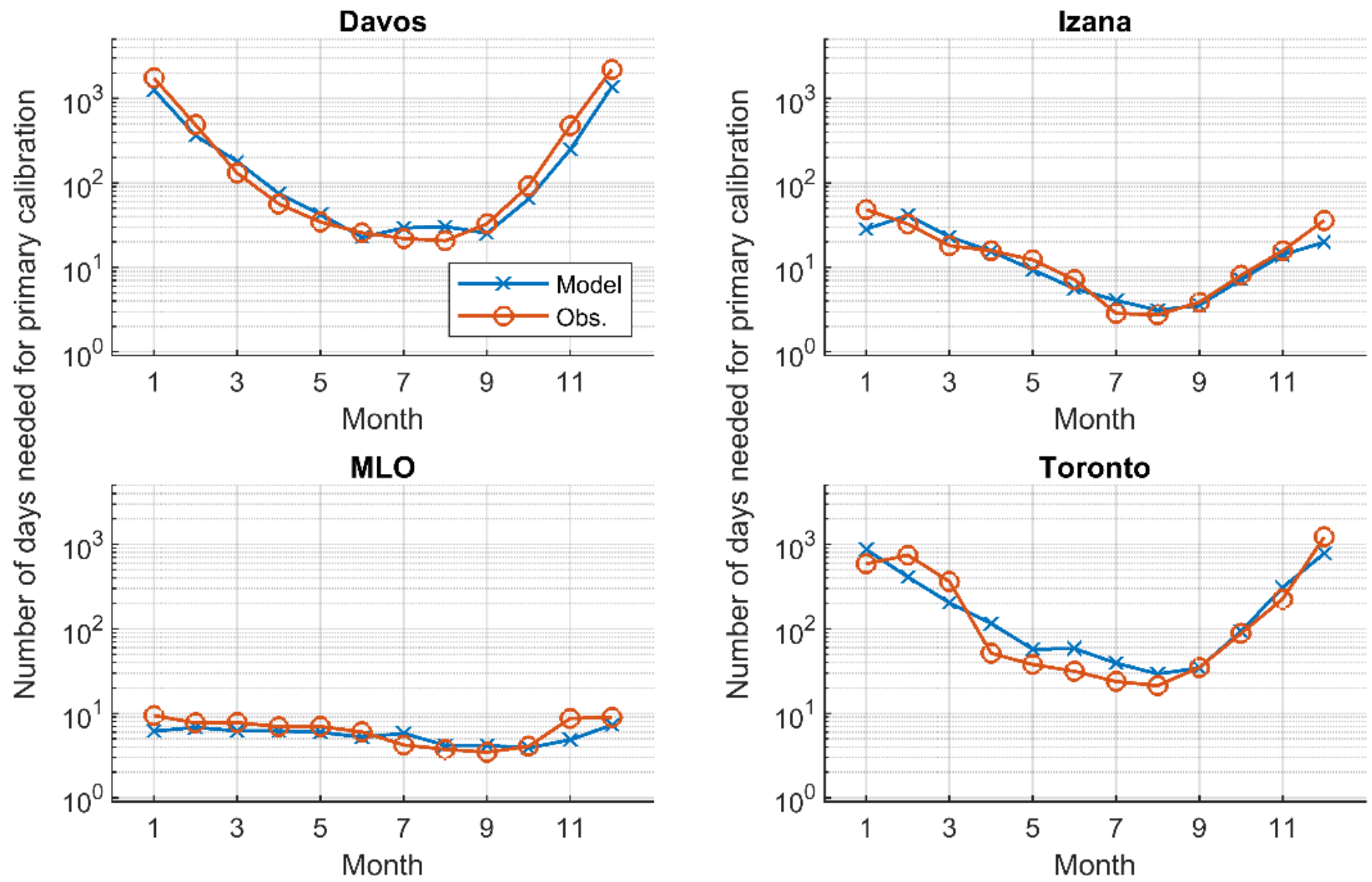

**Figure 7. The number of days needed for primary calibration work at different sites in different months. Blue lines are the results from the ETC simulation model based on MERRA-2 reanalysis, red lines are the results using real Brewer observations. Both blue and red lines use the same observation records from the CIMEL sunphotometer and Brewer DS ozone to infer aerosol and weather conditions.**

## 5 Applications for individual sites

Figure 7 demonstrates that successful calibrations at MLO require less than 10 days in any month, but the best time for calibration is August–October. This result is in excellent agreement with the findings of Gröbner and Kerr (2001) and Kerr (2010). During these months, four days are typically enough to achieve the required ETC uncertainty of 5 R6 units. This is in line with the ECCC calibration strategies in the last few decades. The ECCC world reference instruments were regularly calibrated at MLO about every two years since the beginning of the Canadian Brewer ozone monitoring program in the 1970s (Kerr, 1997; Kerr et al., 1998; Fioletov et al., 2005). Details of dates and instruments for recent MLO calibration trips were provided by Zhao et al. (2021). Typically, these calibration trips were carried out in October (no later than November), when short-term ozone stability and weather conditions were acceptable at the site. For each calibration trip, one of the world reference instruments performs about two weeks of observations at MLO (Kerr, 2010).

For the Izaña site, the AOD conditions in July would have an impact on the primary calibration work. Also, Izaña's calibration conditions show a stronger seasonal dependence than MLO (see Fig. 7). As a result, August to September is the best time window to perform primary calibrations at this site, which can guarantee a higher chance to obtain good-quality ETCs that are less affected by short-term ozone variability and dust events. Different from the ECCC Brewer team who travels to MLO with

world reference instruments, the AEMET Brewer team has the regional reference instruments perform year-round observations at Izaña (except for field campaigns that require the move of the travelling instrument). Thus, these regional reference instruments routinely utilize the PCM to monitor their long-term stability at this site.

Although the Brewer instruments at Davos or Toronto could not easily obtain their $ETCO_3$ via PCM, their long-term individual ETCs still provide key information that can assist in monitoring and assessing their performance. For example, Brewer #163 is a good example to show that its long-term individual ETCs follow very well with its $ETCO_3$ received via CTM (i.e., see the agreement of red and green lines in Fig. 2). Such information can help the local Brewer operation team to be aware of any potential system changes to (1) call for investigations and services and (2) make necessary corrections or calculate new $ETCO_3$ for the field instruments. In practice, by averaging individual ETCs of a field Brewer instrument, which are generated from warmer months (e.g., for Davos or Toronto, from May to October), one can generate its yearly "statistical $ETCO_3$" for comparison with $ETCO_3$ reported in ICF (i.e., via CTM) to verify the stability of the Brewer. Thus, we recommend that data from field Brewers be periodically analyzed using the PCM for long-term data quality assurance.

## 6 Conclusion

This work assessed the calibration conditions for four Brewer observation sites, which include sites of the Brewer world and European regional reference triads. The primary calibrations performed at different sites are found to be affected by many site-specified factors, including short-term ozone variability, aerosol, and cloud conditions. Among these factors, the short-term ozone stability has the strongest impact on the calibration uncertainty (i.e.,10–20 R6 units for MLO and Izaña in summer months, and increase to more than 200 R6 units for Davos or Toronto in winter months; without consider specific instrumental stability and characteristics). This uncertainty is due to the fact that Langley plot techniques used for Brewer primary calibrations are all based on the same assumption — the TCO is not varying during a calibration session (i.e., half-day period). Although the cause of the uncertainty is clear, in practice, it is difficult to identify and remove spurious low-quality ETCs (the ones derived from observations with large short-term ozone variation conditions, e.g., half-day TCO changes > 2 DU). This is simply due to the Brewer spectrophotometer taking the most accurate TCO observations among ground-based instruments (precision within 1%, corresponding to about 3–4 DU in typical TCO conditions). I.e., it is very challenging to provide independent high-quality TCO data that can prove the ozone field was or was not stable for a half-day period. Currently, other ground-based instruments, satellites, or reanalysis models (such as MERRA-2 used here) do not have sufficient accuracy and precision to support Brewer's primary calibration work.

To lower the uncertainty of the calibration product (to meet the calibration goal; $ETCO_3$ with an uncertainty of ±5 R6 units), a minimum number of individual ETCs needs to be averaged, which depends on the season and location of the calibration site.

Utilizing short-term ozone stability conditions estimated by using long-term real observations and the new ETC simulation model, this work presented the site-specified criteria for the Brewer primary calibration work. For MLO and Izaña, when accounting for aerosol and weather conditions, typically, around ten days of observations are needed from August to September. The primary calibration work still can be performed in other months at these sites, particularly at MLO; however, it will be at the cost of more observations being needed and with a higher probability of encountering poor calibration conditions (mainly the days with higher short-term ozone variability).

Using the findings of the site-specified calibration conditions, the optimized primary calibration strategies of the Brewer world and European regional reference triad are described in this work. These unique strategies are designed to ensure primary calibration results from MLO and Izaña can meet the same high-quality standard. Although the PCM can not deliver high-quality $ETCO_3$ from mid-latitude sites, such as Davos and Toronto, this technique can be used as a tool to assist local operations, long-term monitoring, and data assurance work.

The ETC simulation model developed in this work is another important tool that can be used to provide a quick guide for the evaluation of any Brewer field site's conditions. It is worth noting that the model is not dependent on real Brewer observations but can provide key information for any location. In future, a similar model can be developed to improve our understanding of Brewer calibration transfer work, e.g., at a given site, how many observations are needed to meet the requirement of calibration transfer and which months are best to perform such actives. This research can be carried out in the future to further improve our knowledge of uncertainties of Brewer TCO observations, calibration, and calibration transfers.

**Appendix A**

As we are using MERRA-2 TCO to generate the "true" ozone daily variation trend used in the ETC simulation model, it is necessary to verify MERRA-2 TCO's performance at those selected Brewer observation sites. Note that if the reanalysis TCO has a major difference compared to observations, then the ETC simulation model's results could have bias. For example, if MERRA-2 TCO has a large difference compared to Brewer observations in terms of seasonal or diurnal variations, our simulated individual ETCs would also be affected and biased.

To evaluate MERRA-2 TCO, Brewer TCO observations are averaged into hourly mean to match MERRA-2 temporal resolution. Note that we are directly using MERRA-2 TCO values from the model grids that cover those Brewer sites. Figure A1 shows the timeseries of the coincident Brewer and MERRA-2 TCO dataset, where the "shading areas" are individual hourly values (gray dots for Brewer, light-orange crosses for MERRA-2), and the red and blue lines are the monthly mean of Brewer

and MERRA-2 TCO, respectively. Overall MERRA-2 TCO follows the observed TCO seasonal variation very well with systematic bias within ±2.5%. Figures A2 and A3 show the percentage difference of Brewer and MERRA-2 TCO data binned by months and hours of local standard time. The error bars in the figures show 1σ values of the difference. These results confirm that MERRA-2 TCO data has a slightly higher positive bias in Davos and Toronto, while negative bias is found in Izaña and MLO. However, the overall percentage difference is within ±5% on the 1σ level. These small differences in seasonal and diurnal patterns should only have a limited impact on the modelled daily ozone variation pattern or the ETC simulation model. This is also confirmed by the acceptable agreement found between the model and observation-based calibration site condition results (e.g., see Fig. 7).

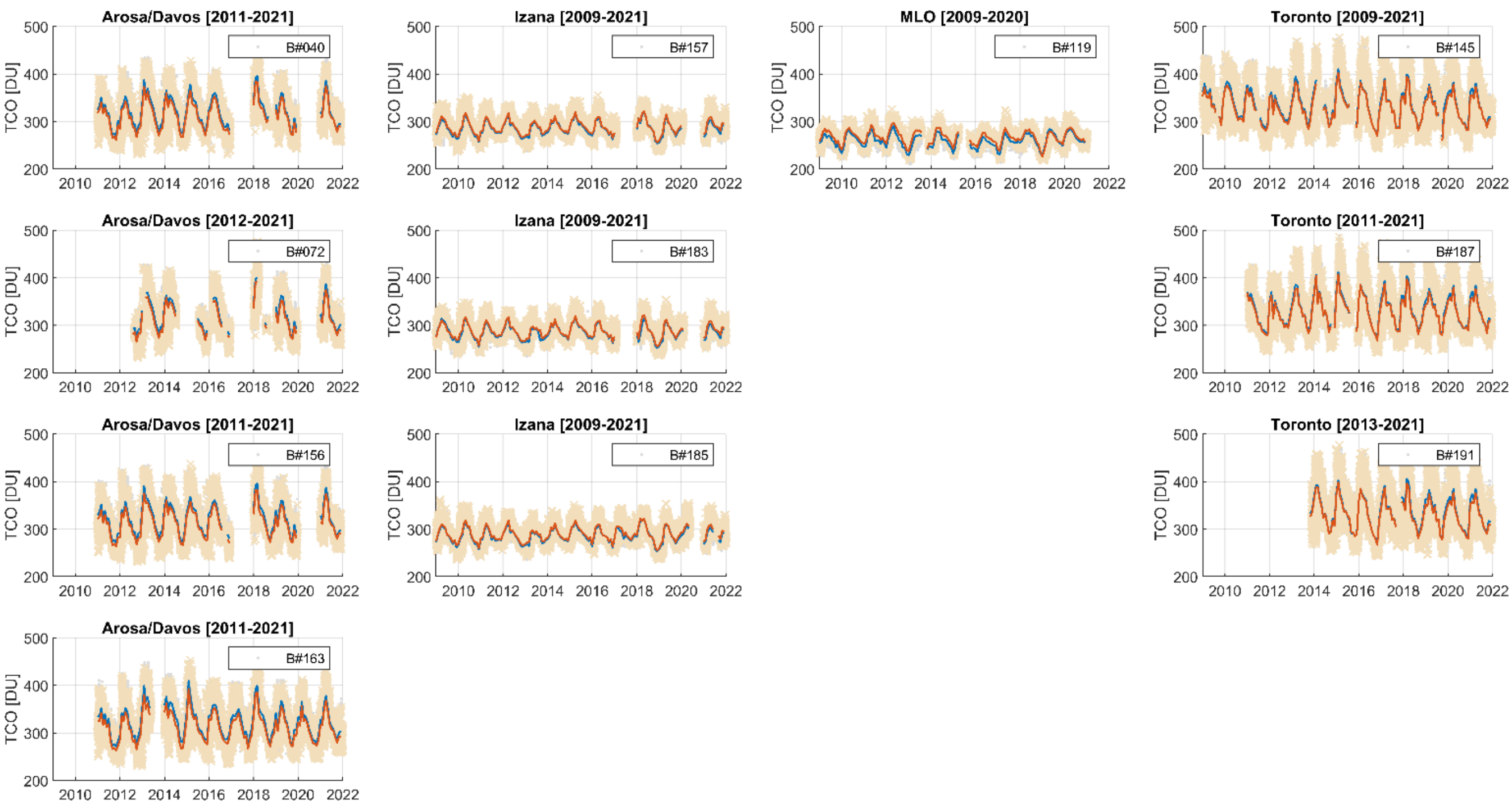

**Figure A1. Timeserise of Brewer and MERRA-2 TCO. The red and blue lines are the monthly mean of Brewer and MERRA-2 TCO, respectively, the "shading areas" represent individual hourly values. The Brewer instrument numbers are given in the legend.**

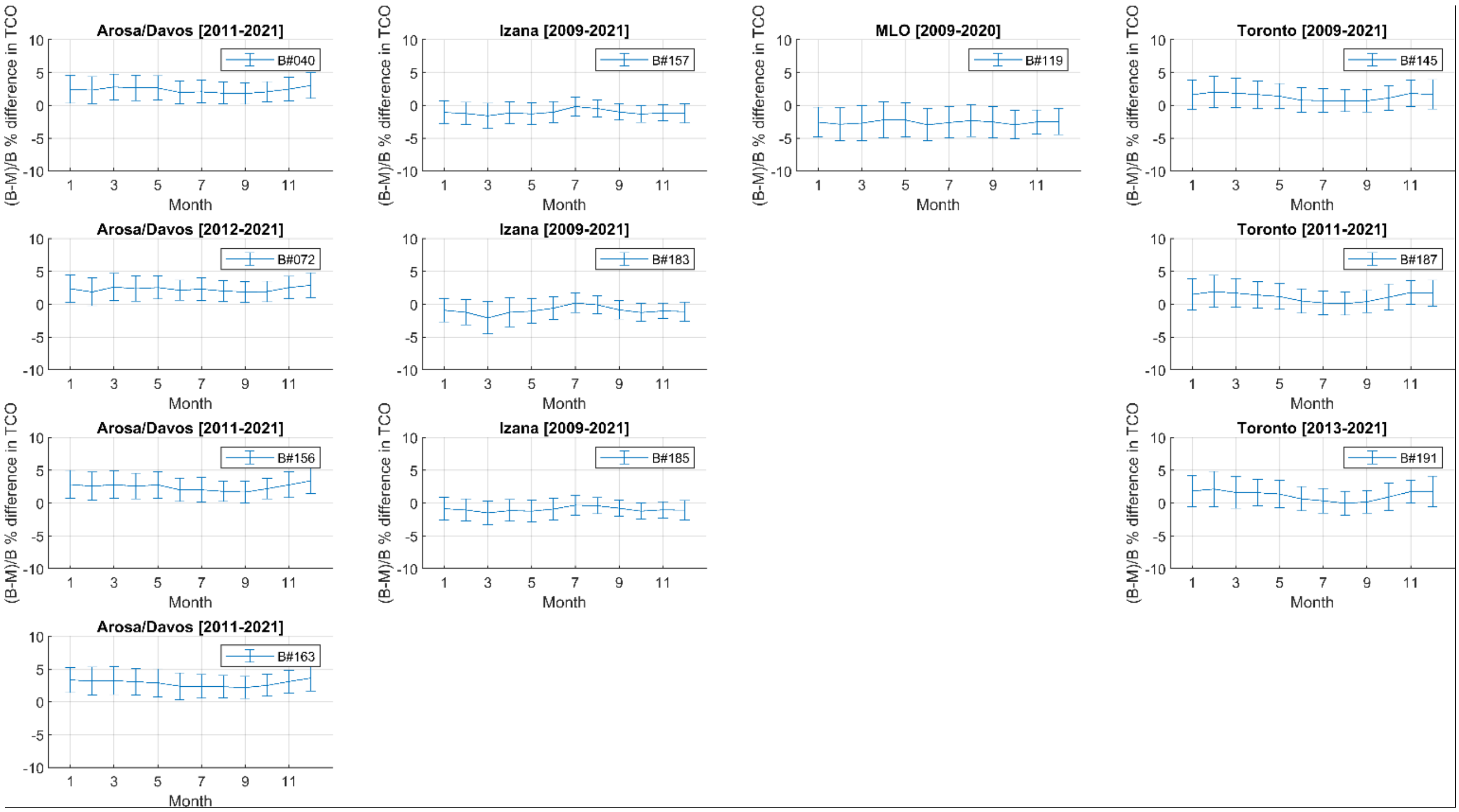

**Figure A2. Brewer and MERRA-2 TCO % difference binned by month. The percentage difference is calculated as (Brewer – MERRA-2)/Brewer. Error bars are the 1σ of the monthly percentage difference. The site and data period are shown in the title of each panel. The Brewer instrument numbers are given in the legend.**

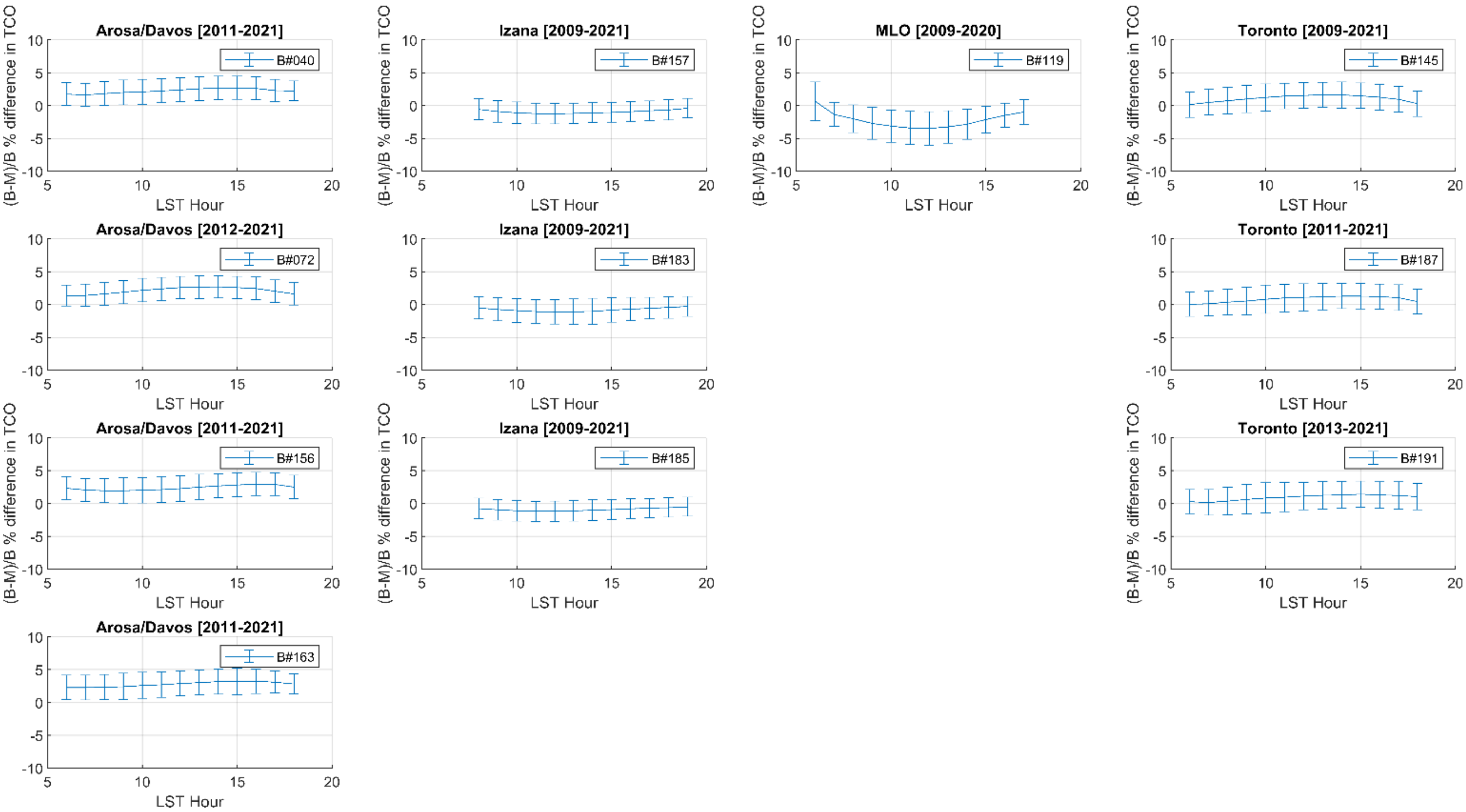

**Figure A3. Brewer and MERRA-2 TCO % difference binned by the hour of local standard time. The percentage difference is calculated as (Brewer – MERRA-2)/Brewer. Error bars are the 1σ of the hourly-binned percentage difference. The site and data period are shown in the title of each panel. The Brewer instrument numbers are given in the legend.**

## Appendix B

The monthly standard deviations of individual ETCs from all 11 instruments are shown in Figure B1. Due to instrumental factors and data gaps, the standard deviations from the different instruments at the given site are not identical. However, the general seasonal pattern from all instruments at the four sites is consistent with the results from the ETC simulation model. Similarly to the modelled results, the real observation data fitted individual ETCs are less scattered in lower latitudes and in summer seasons.

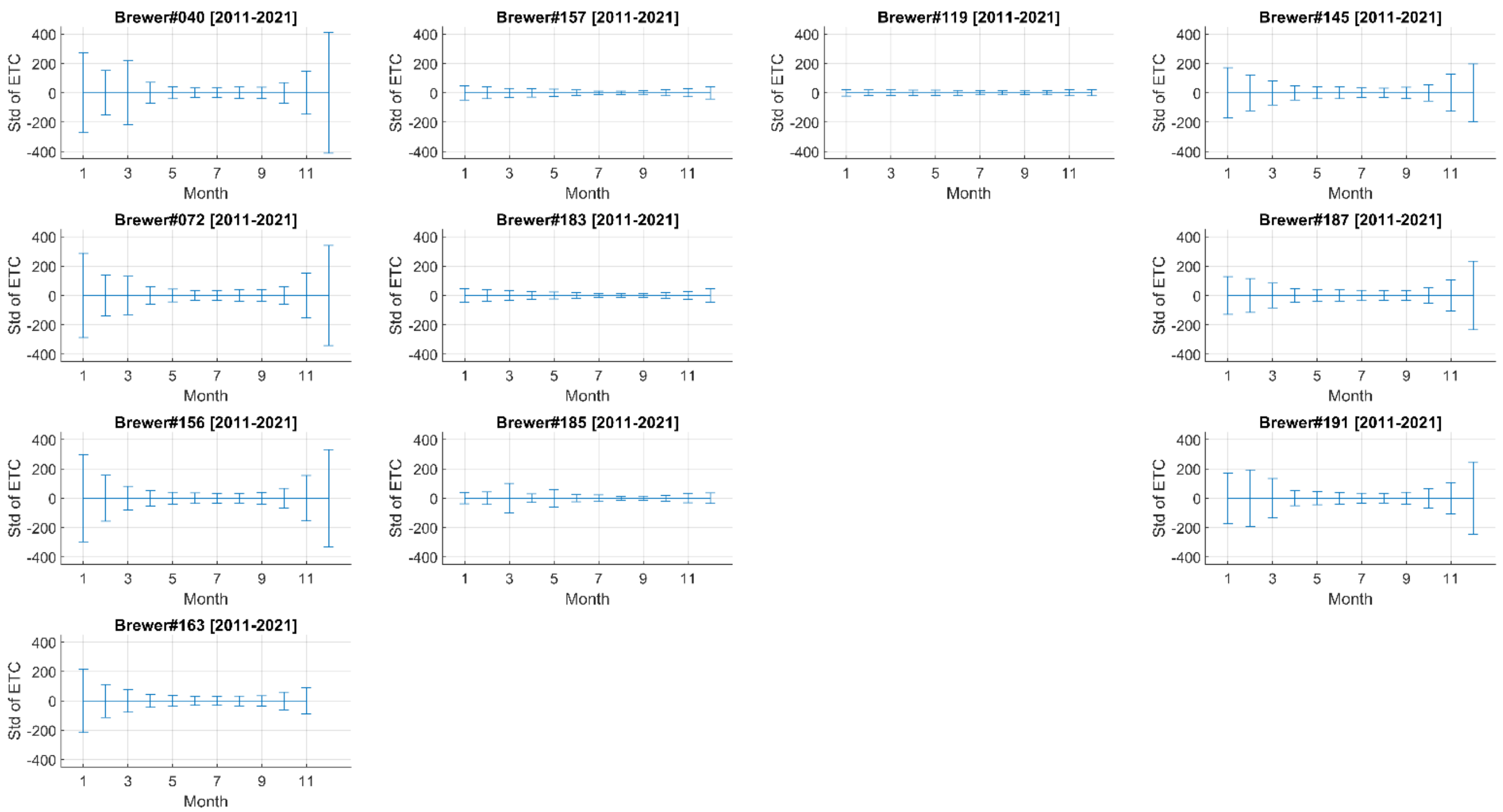

**Figure B1. Monthly standard deviations of individual ETCs for all Brewer instruments included. Error bars represent the 1σ of individual ETCs for the month. The instrument serial number and its observation period are shown on the title of each panel.**

*Data availability*. Brewer data are available from WOUDC (http://woudc.org/, last access: 1 November 2022) and EUBREWNET (https://eubrewnet.aemet.es/, last access: 1 November 2022). Cimel data are available from AERONET (https://aeronet.gsfc.nasa.gov/, last access: 1 November 2022). MERRA-2 TCO data are available from NASA EARTHDATA (https://disc.gsfc.nasa.gov/datasets/M2T1NXSLV_5.12.4/summary?keywords=M2T1NXSLV%20v5.12.4, last access: 1 November 2022). Any additional data may be obtained from Xiaoyi Zhao (xiaoyi.zhao@ec.gc.ca).

*Author contributions.* XZ analyzed the data and prepared the manuscript, with significant conceptual input from VF and JK. XZ had critical feedback and discussions from all co-authors. MB, RS, VF, XZ, and SCL operated and managed the Canadian Brewer Spectrophotometer Network. IA, AO, and MB prepared the Canadian Brewer dataset. AR, JLS and AB managed and operated the RBCC-E; AR is the PI of Brewer#157, #183, and #185. JG is primarily responsible for hosting the Swiss Brewer triad at Davos and is the PI of Brewer #163. EMB is the PI of the Swiss Brewer triad (Brewer #040, #072, and #156). LE and JLS prepared Swiss and Spanish datasets and assisted in the data analysis. HS organized the joint Brewer Central Calibration Laboratory meetings and facilitated the research project.

*Competing interests.* The contact author has declared that neither of the authors has any competing interests.

*Acknowledgements.* We gratefully thank the National Oceanic and Atmospheric Administration (NOAA) and Mauna Loa Observatory staff for supporting Canadian Brewer operation and calibration work at MLO. We thank NASA AERONET for providing Cimel Sunphotometer aerosol data. We thank the Global Modeling and Assimilation Office (GMAO) for providing MERRA-2 data. XZ thanks Volodya Savastiouk for his valuable comments and suggestion for this work. XZ thanks Thomas F. Eck from NASA for providing suggestions on Cimel Sunphotometer data interpretation.

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
