# Peer review of "The site-specific primary calibration conditions for the Brewer spectrophotometer"

_Atmospheric Measurement Techniques, 2023_

## Author Comment (AC1)

**Response to Referee #1:**

We thank referee #1 for their very helpful comments. Our responses are given below in black with the referee's comments in blue. The new text in the modified manuscript is given in red (italicized).

GENERAL COMMENTS

The manuscript by Zhao et al. analyzes the atmospheric conditions affecting the primary calibration of reference Brewers at four sites, with particular reference to the short-term variations of the total ozone column. A modelling framework is developed to simulate the effect of the ozone variations on the extraterrestrial calibration factor, using MERRA-2 reanalyses as input. The study aims at assessing: "(1) why Brewer primary calibration work can only be performed at certain sites... and (2) what is needed to assure the equivalence of calibration quality from different sites" (lines 104-106). While the answer to the first question is rather obvious, the second research question is very relevant to the ozone science. The paper is generally written in a clear way. Based on these considerations, I would recommend publication of the manuscript after some corrections.

We appreciate referee #1 for this very positive feedback.

**SPECIFIC COMMENTS**

- Structure. It may be a matter of taste, but I would recommendation to follow a more traditional paper structure better highlighting the methods, the results and the discussion. I feel a bit confused, for example, when reading that Sect. 3 is titled "Primary calibration" and Sect. 3.2 is titled "Primary calibration method": what difference should the reader expect from the two sections?

We thank referee #1 point this out. The main difference is Sect. 3.1 only provides information on how to do one Langley plot, while Sect. 3.2 is about why and how to "combine" multiple Langley plots to achieve the calibration goal. Following the suggestion, we renamed the Sect. 3.2 from "Primary calibration method" to "ETC product".

- Different effects of short-term ozone variability. I am sure that the authors can introduce this topic in a more clear and tidy way (lines 320-330). Indeed, daily variations in ozone and instrumental factors can result in both random "noise" in the ETC determinations (appropriately tackled with type-A evaluation of the uncertainty) and systematic effects, notably in presence of recurring daily or sub-daily patterns (e.g., photochemistry? local pollution? instrumental artifacts?). I think that

this distinction requires further discussion, especially if the authors examine the linear and parabolic terms of the daily variations (e.g., how regularly these patterns occur?).

Following the suggestion, we provided more analysis of the liner and parabolic terms of the daily variations. The answer to this question is also related to the next one. So, we merged these replies together (i.e., see more detailed answers for the next question). We also want to point out there were comparisons made between Brewer #156 and QASUME (which was calibrated in the laboratory). Egli et al. (2022) shows that the difference between the lab calibrated approach (which is insensitive to systematic ozone variations that potentially affect the Langley-procedure) and the Langley-plot based approach is less than 1%.

> *An alternate approach has been demonstrated by Egli et al. (2022) using the QASUME spectroradiometer calibrated in the laboratory with SI-traceable radiation standards to retrieve the atmospheric TCO from direct spectral solar irradiance measurements without requiring an in-situ based Langley calibration, and thereby being insensitive to possible systematic ozone variations that potentially affect the Langley-plot based calibration. Collocated measurements between QASUME and Brewer #156 have shown good agreement in retrieved total column ozone with less than 1% difference.*

- Relation between simulated and observed ETCs and reliability of MERRA-2 reanalysis. The authors state that:

1. the "large day-to-day difference is mainly due to short-term ozone variability" (l. 297)
2. they use MERRA-2 to "isolate the short-term ozone variability impacts" on ETCs (l. 360 - what does "isolate" mean, exactly?)

We have revised the sentence to make it clearer.

> *To avoid such instrument-related issues and meteorological factors, MERRA-2 reanalysis ozone data are used in an ETC simulation model developed to isolate the short-term ozone variability impacts (i.e., to exclude other instrumental and natural factors that could affect ETC) and also to avoid data gaps.*

3. "the impact of linear and quadratic ozone variations can be independently assessed" (l. 405)
4. the uncertainty of MERRA-2 reanalysis is still too high to be used for correcting the single ETCs for short-term ozone changes (lines 600-603).

All things considered, a straight question is: is there a direct correlation between the ozone variations from MERRA-2 (e.g., in terms of the "b" or "c" coefficient) and the individual observed ETCs? Or should simulations using MERRA-2 data (and their good agreement with the observed behaviour, e.g. in Figs. 5e-h) only be interpreted as an average indication of the effect and its magnitude, depending on the site and the considered season? Can the authors further elaborate on the results reported at lines 406-408?

We thank the referee for this important question. So, we examined the agreement of MERRA-2 and Brewer observations in terms of their fitted daily 2nd-order polynomial lines (TCO = $a+b\Delta t+c\Delta t^2$). Figure R1 is an example of the results for Br#191, which shows the distribution of each fitting coefficient (blue bars are histograms of MERRA-2 fitting coefficients, while brown bars are histograms of Brewer #191's fitting coefficients). The values indicated in the legends are the mean ± 1 sigma of the coefficients. Figure R1 shows that, statistically (on average), the ozone daily variations from MERRA-2 generally agreed with Brewer's observations (this is also supported by Fig. A3, which shows their % difference binned by the hour of local standard time), i.e., we did not see any strong evidence that MERRA-2 daily variation deviated from the observations.

[Figure]

**Figure R1. Histograms of 2nd-order polynomial fitting coefficients for daily ozone (for MERRA-2 and Brewer #191 at Toronto). The values indicated in the legends are the mean ± 1 sigma of the coefficients.**

As shown in the figure below (Fig. R2), we also examined the correlation between MERRA-2 and Brewer's fitted daily curves via their correlation coefficient (see blue bars). Here, we have >50% of cases the reanalysis and measurements show a strong correlation (>0.8). Note that for days with low ozone changes in a day (e.g., the ozone field is stable), the correlation between MERRA-2 and Brewer's fitted daily curve would be expected to have a relatively low correlation. Thus, if we further remove the days where ozone change in a day is less than median values, we have almost 80% of the dataset has $R^* > 0.8$. This means when we have moderate ozone daily changes (above median value), for about 80% of the time, MERRA-2 can represent >80% of ozone changes observed by Brewer in those days. Please note that, regarding the portion of the data that has been excluded in the $R^*$ calculation (due to low ozone changes in a day), it also means both MERRA-2 and Brewer agree on those days that the ozone was relatively stable.

[Figure]

**Figure R2. Histogram of correlation coefficients between MERRA-2 and Brewer observations. R is the correlation coefficient for the entire coincident dataset; $R^*$ is the correlation coefficient for the coincident data that has $\Delta O_3 \geq$ median of daily changes.**

Further analysis of the differences between the fitted daily curves is shown below (Fig. R3). Here, for each day, the histograms of the difference between MERRA-2 and Brewer fitted $2^{nd}$ polynomial curves are examined. The middle panel of Fig. R3 shows the standard deviation of the difference (MERRA-2 - Brewer), where the median value is only 3 DU. This result indicates that, for most conditions (in a given day), the difference between MERRA-2 and Brewer ozone is 3 DU on one sigma level. The distribution of the mean and median of the difference is not strongly deviated from the Gaussian shape (see the first and the last panel in Fig. R3). Thus, statistically, the MERRA-2 ozone followed a similar pattern as observed ozone (with some minor offset for this site (Toronto), which was also reported in our other MERRA-2 vs. Brewer analysis in Appendix A).

[Figure]

**Figure R3. Histogram of the difference between MERRA-2 and Brewer fitted ozone curves.**

Figures R4–6 show the same analysis as Fig. R1, but for all 11 Brewers included in this work. Figure R4 is the summary of fitted constant term for all 11 instruments; Fig. R5 is the summary of fitted linear term, and Fig. R6 is the one of fitted quadratic term.

[Figure]

**Figure R4. Histograms of 2ⁿᵈ order polynomial fitting coefficient (0 order term; constant term) for daily ozone. The values indicated in the legends are the mean ± 1 sigma of the coefficient.**

[Figure]

**Figure R5. Histograms of 2$^{nd}$ order polynomial fitting coefficient (1$^{st}$ order term; linear term) for daily ozone. The values indicated in the legends are the mean ± 1 sigma of the coefficient.**

[Figure]

**Figure R6. Histograms of 2$^{nd}$ order polynomial fitting coefficient (2$^{nd}$ order term; quadratic term) for daily ozone. The values indicated in the legends are the mean ± 1 sigma of the coefficient.**

However, we admit that the agreement between reanalysis and observation is not perfect. There are many factors that could contribute to the difference, which include but are not limited to, such as model resolution, line-of-sight of Brewer instrument, vertical sensitivity difference, etc. However, MERRA-2 is good enough for the current modelling work to prove that ozone variability in the short-term is the major challenge in Brewer primary calibration work. Figures R4–6 have been added to Appendix A as Figs. A4–6, and some of these discussions are included in the manuscript (Sections 3.3, 6, and the Appendix).

*Simple TCO variation amplitudes (i.e., maximum value subtract minimum value for that given period) for whole daytime, a.m., and p.m. sessions are reported in Fig. 3d to f, with the probability of variation amplitudes less than 1.5 DU (suitable conditions for Brewer Langley calibration work) shown in the legends. For example, the results (Fig. 3e) show that Brewers at MLO and Izaña have a better chance (e.g., 55% and 45% for morning sessions, respectively) to have a good stable short-term ozone field to produce high-quality individual ETCs via Langley plot techniques than the ones at Davos (23%) or Toronto (20%). Statistics of MERRA-2 and Brewer's fitted daily ozone variations are provided in Appendix A (see Figs. A4–6).*

*Currently, other ground-based instruments, satellites, or reanalysis models (such as MERRA-2 used here) do not have sufficient accuracy and precision to support Brewer's primary calibration work. In general, MERRA-2 can capture the general pattern of ozone daily changes. The median value of the standard deviation of the difference between MERRA-2 and Brewer ozone is only 3 DU (about 1%). However, limited by many factors (such as spatial and temporal model resolution, instrument line-of-sight, etc.), currently, the modelled ozone could not perfectly reproduce the observations record.*

*These small differences in seasonal and diurnal patterns should only have a limited impact on the modelled daily ozone variation pattern or the ETC simulation model. This is also confirmed by the acceptable agreement found between the model and observation-based calibration site condition results (e.g., see Fig. 7). We also examined the agreement of MERRA-2 and Brewer observations in terms of their fitted daily 2$^{nd}$-order polynomial lines (TCO = a+bΔt+cΔt$^2$). The histograms of their fitted coefficients are shown in Figs. A4–6. Statistically, MERRA-2 can follow the local daily ozone variation pattern (the fitting terms by using MERRA-2 and Brewer observations are agreed on a 1-sigma level).*

- Sect. 4: is SO2 an interfering factor at MLO?

We thank the referee to point out this special condition factor for the MLO site. Yes, theoretically, $SO_2$ could affect the calibration of Brewer. However, we also must point out that the wavelengths of Brewer's operational ozone algorithm (nominal values at 310.1, 313.5, 316.8, and 320 nm; the four slits with the longest wavelengths measured by Brewer) were selected to be located at wavelengths that eliminate differential absorption due to $SO_2$ (Kerr, 2010). Figure 6 of Evans et al. (1981) shows measurements of large values of $SO_2$ (>~50 DU) during the passage of volcanic debris from Mount St. Helen which erupted in late May, 1980. These measurements were made with the Brewer Mark I version. There is no suggestion that $SO_2$ impacts the ozone measurement, although it is difficult to tell since ozone itself is changing. Figure 4 of Kerr et al., (1985) shows another major $SO_2$ pollution event (~40 DU) in August, 1983 observed by

two Mark II instruments (#5 and #8). In this case both instruments show no significant impact by SO$_2$ on the ozone measurements which remained relatively stable during this event.

In addition, the observatory (MLO) is 2000 m above the volcano, so degassing SO$_2$ from Kilauea volcano typically cannot reach the observatory. In fact, for the world reference instruments, we rarely detected any SO$_2$ signal during the previous calibration trips also because typically prevailing winds (from east to west) direct SO$_2$ plumes away from the observatory. Figure R7 is the histogram of SO$_2$ column measured by Brewer #119 at MLO. As Fig. 7 shows, SO$_2$ at MLO is typically within the noise level, and even 5 DU SO$_2$ is very rare for MLO.

[Figure]

**Figure R7. Histogram of column SO$_2$ measured by Brewer #119 at MLO.**

In 2018, the Kilauea volcano eruption occurred and was still moderately active in 2019 while the ECCC Brewer team was there doing calibrations. When the winds became more southerly, the instrument actually saw SO$_2$ values reach as high as 10 DU. Such data are in fact good for the ETCSO2 calculation. To further demonstrate if presence of high SO$_2$ could impact ETCO3, we also plotted the fitted ETCs along with the high SO$_2$ observed for 2016 and 2018 cases (note these are rare cases, and were not the periods that Brewer world reference instruments were performing calibration at MLO). Figure R8 shows that, even when SO2 column > 20 DU, there is no indication that the fitted ETCO3 values are driven by the SO$_2$ values.

[Figure]

This information has been included in the revised manuscript (Sect. 4.2 and Appendix B).

*Meanwhile, due to the different characteristics of each site, different optimized strategies must be selected by the AEMET and ECCC Brewer teams in practice. For example, it was found the MLO site has more cloudy conditions during afternoon sessions and thus, more observations must be made to ensure a good balance of morning and afternoon sessions. Another factor that might impact primary calibration at the MLO site is the $SO_2$ plume from the volcano. However, we must point out that the wavelengths of the Brewer operational ozone algorithm were selected to be located at wavelengths that eliminate differential absorption due to $SO_2$ (Kerr, 2010). Previous work shows no evidence that $SO_2$ will affect Brewer ozone observations (e.g., Evans et al., 1981; Kerr et al., 1985). Examples of fitted ETC values with observed $SO_2$ are shown in Appendix B.*

*The MLO is 2000 m above the Kilauea volcano, and thus degassing $SO_2$ from the volcano typically cannot reach the observatory. In addition, with the world reference instruments, we rarely detected any $SO_2$ signal during previous calibration trips because typically prevailing winds (from east to west) direct $SO_2$ plumes away from the observatory. Figure B2 is the histogram of $SO_2$ column measured by Brewer #119 at MLO. As Fig. B2 shows, $SO_2$ at MLO is typically within the noise level, and even 5 DU $SO_2$ is very rare for MLO. The example timeseries from 2016 and 2018 demonstrated that the fitted ETC values are not driven by the variation of $SO_2$, even when $SO_2$ column values > 20 DU.*

[Figure]

*Figure B2. (left) Histogram of column SO₂ measured by Brewer #119 at MLO (middle and right). Example timeseries of fitted ETCs and observed SO₂.*

**TECNICAL REMARKS**

- Title: I'm not a native English speaker, but "site-specified" sounds a bit odd (should it be "site-specific"?)

Done.

- l. 29, "stable short-term ozone field": "stable ozone field in the short term"

Done.

- l. 36: notice that if R6 units are divided by the differential absorption coefficient, the same quantity can be expressed in DU (relative to airmass 1). The equivalent quantity in DU can thus be specified in the text for greater clarity

Thanks for this good suggestion. However, the differential absorption coefficient ($\Delta\alpha$) for each Brewer is different. The value is determined by wavelengths and resolution of an individual instrument, and also the ozone cross sections (and the effective temperature) that have been selected (e.g., such as using BOp or IUP cross sections) (e.g., Redondas et al., 2014; Gröbner et al., 2021). Thus, if we select a "typical" $\Delta\alpha$ (e.g., 0.34) and use airmass = 2, we can "convert" R6 into DU. For example, 5 R6 units is equal to 1.47 DU (with $\mu = 1$ and $\Delta\alpha = 0.34$). Such conversion provides a clear meaning of the value, however, it will be more difficult for us to quantify the true instrument response alone. As the main idea of this work is to set up and examine the technical standard for Brewer calibration work (e.g., primary calibration to meet the 5 R6 unit goal), we would prefer to continue to use the R6 unit. We have included such information in the revised work to make this clearer for readers.

*For a Brewer instrument, the goal of such calibration is to derive its unique ETC value (that can be used in TCO calculation), with uncertainty within ±5 R6 units (Zhao et al., 2021). Here R6 is a measurement-derived double ratio in the actual Brewer processing algorithm, corresponding to the measured slant column ozone (e.g., Savastiouk, 2006; Zhao et al., 2021). Note that in typical conditions (e.g., $\Delta\alpha = 0.34$ and $\mu = 2$), 5 R6 units is equal to 0.74 DU or about 0.25% for a typical ozone value of 300 DU.*

- l. 117: it should be mentioned that the second "channel" is used for SO2 retrievals

Done.

*The Brewer spectrophotometer is a modified Ebert grating spectrometer that was designed to measure almost simultaneously the intensity of radiation at six UV channels (nominal wavelength at 303.2, 306.3, 310.1, 313.5, 316.8, and 320.1 nm). The first channel is almost exclusively used for wavelength calibration, the second channel is used for SO$_2$ retrieval.*

- l. 124-125: this formula should be written in a distinct line, so that the definition of F can be referenced more easily if needed (e.g., line 225)

Done.

*F, Δα, and Δβ are the linear combinations of the logarithms of the measured intensity, the effective ozone absorption and the Rayleigh scattering coefficients, respectively. For example,*

$$F = -log_{10}(I_3) + 0.5\,log_{10}(I_4) + 2.2\,log_{10}(I_5) - 1.7\,log_{10}(I_6), \qquad (2)$$

*where I$_3$ to I$_6$ are the photon count rates at the last four longer wavelength channels (Kerr et al., 1985).*

*With assumptions that TCO values (Ω) are constant through the calibration session (half-day) and aerosol has negligible impact, the instrument response F$_i$ (see Eqn. 2) adjusted for instrumental (dead time, dark counts) and some atmospheric (Rayleigh scattering) factors is a linear function of airmass (μ$_i$):*

$$F_i = ETC + 10Ω Δα μ_i + e_i \qquad (3)$$

*where, i is the observation number, Δα is the effective ozone absorption, and 10 is a scaling factor used in the Brewer software, ETC is the extraterrestrial constant (here, ETC = $-10^4 × F_0$).*

- l. 125-126, "the last four longer" --> "the four longest"

Done.

- l. 146: notice that Cede et al., 2006 employ MkIII Brewer, which clashes with the premise "NO2, by Mark IV only"

Done.

*The Brewer spectrophotometer provides data products that include column ozone (e.g., Kerr, 2002; Kerr et al., 1981), column sulphur dioxide (SO$_2$; e.g., Fioletov et al., 1998; Zerefos et al., 2017), column nitrogen dioxide (NO$_2$, by Mark III and IV; e.g., Kerr et al., 1988; Cede et al., 2006; Diémoz et al., 2021), spectral UV radiation (e.g., Bais et al., 1996; Fioletov et al., 2002), aerosol optical depth (AOD) (e.g., Kazadzis et al., 2005; Marenco et al., 2002; Diémoz et al., 2016; López-Solano et al., 2018), and effective ozone layer temperature (Kerr, 2002).*

- l. 168: "wavelength" --> "wavelengths"

Done.

- l. 191-195: isn't it a repetition of what is already said in the Introduction?

We removed the repeated part.

> *Four long-term Brewer calibration and/or operation sites are included in this work: Arosa/Davos, Switzerland; Izaña, Spain; MLO, Hawaii, U.S.A; and Toronto, Canada (see Table 1 for details).  (Stübi et al., 2017) Arosa/Davos, Izaña, and Toronto are the operation sites for the Swiss triad, RBCC-E, and the world references, respectively.*

- Eq. (2): notice that the equation is different from (1). In addition to the 10 factor, the sign of the ozone term is opposite. Please, use only one convention for the sign of the differential coefficient (e.g., define it positive). Also, use either the "F0" or "ETC" expression

We thank the referee to point out this issue, and we apologize for the confusion. The main difference between Eqn. (1) and Eqn. (2) (in the revised paper, corresponding to Eqn. (3)) is the definition of instrument response. In the original definition (e.g., Kerr, 2010), the instrument response ($F$) is a negative number; for convenience (and due to the storage limit in old days), the intensities ($I$) measured by the instrument were stored in $10^4\log_{10}(I)$ scale. The ETC is converted to a positive number via $ETC = -10^4 \times F_0$ (it was provided in Line 231 in the original manuscript). The factor of 10 in Eqn. (2) was for the fact that the Brewer algorithm has this $-10^4$ factor included and also we converted the total ozone from the unit of cm to the Dobson Unit ($1\ DU = 10^{-3}\ cm$). To make the equations here to be more consistent and clearer, we modified the Eqns. 1–2, and the conversion of ETC from $F_0$ accordingly (i.e., re-defined the $F$ values to be positive). We also provided a simple explanation of factor 10 in the text.

> *The four longer wavelengths are used for the TCO ($\Omega$) retrieval via the following equation:*
>
> $$F - \Delta\beta \cdot m = F_0 + \Delta\alpha \cdot \Omega \cdot \mu \qquad (1)$$
>
> *where, m and μ are the enhancement factors for the slant-to-vertical path length of the direct radiation for air and the ozone layer respectively (also known as the air mass factors).*
>
> *With assumptions that TCO values ($\Omega$) are constant through the calibration session (half-day) and aerosol has negligible impact, the instrument response $F_i$ (see Eqn. 2) adjusted for instrumental (dead time, dark counts) and some atmospheric (Rayleigh scattering) factors is a linear function of airmass ($\mu_i$):*
>
> $$F_i = ETC + 10\Omega\Delta\alpha\mu_i + e_i \qquad (3)$$

[revised manuscript text omitted]

---

## Author Comment (AC2)

**Response to Referee #2:**

We thank referee #2 for their very helpful comments. Our responses are given below in black with the referee's comments in blue. The new text in the modified manuscript is given in red (italicized).

**General comments:**

The manuscript "The site-specified primary calibration conditions for the Brewer spectrophotometer" by Xiaoyi Zhao et al., aimed to answer two primary questions related to calibration of the Brewer spectrophotometers. The first question addresses site-specific factors, i.e., why the calibration procedure will not work well at certain locations. The second question, closely-related to the first question, deals with the required conditions to achieve a certain calibration quality.

The answers to these questions are important to the assessment of the measurement quality of the ground-based Brewer (and Dobson) measurement networks, and therefore important to Ozone research. The use of available auxiliary data and the development of a modelling framework using MERRA-2 to answer these questions are done in an innovative and convincing way.

The manuscript is well written and well presented. The methods employed are scientifically robust and they are clearly explained. The figures are clearly illustrated, except for one or two that can be easily improved. This manuscript fits within the scope of AMT. Therefore, I recommend its publication after addressing the comments of Reviewer #1 and some of the minor comments below.

We appreciate the very positive comments from the referee on our work.

**Specific comments:**

1.  The Authors briefly mentioned the ETCs from the ICF (Instrument Constant File). I think it should be clarified in a few words what this is, and how the ETC values in this file are obtained.

We have included a more detailed description as suggested.

> *The documented ETC values from the instrument calibration file (ICF) are shown as green lines with their validation period indicated by vertical black dash lines.* *Note that these ICF ETCs are the numbers used in each Brewer's ozone data production. Here, the world and regional references instruments' ICF ETCs were acquired via PCM (e.g., Kerr, 1997), while the other instruments' ICF ETCs were acquired via CTM during calibration campaigns (e.g., Redondas et al., 2018a).*

2.  Related to comment #1 above, it is my understanding that Brewer spectrophotometers have internal quartz-halogen lamps, which are used for the purpose of monitoring instrument stability and changes in ETC values. It would have been interesting to know the results from these regular lamp tests and how the

lamp test results can be employed to support the calibration method. I would have preferred if there was a discussion about it.

We thank the referee to point out this technical detail, which most non-expert readers would not find. We avoided such details intentionally to make the paper a bit more easily to be absorbed.

The internal halogen lamp test is the SL (standard lamp) test. During such a test, the micrometer position is kept at its operating position for ozone/$SO_2$ measurements, and the intensities at all six wavelengths are recorded by observing with the lamp's light (not the solar light). The same instrument responses (lamp response $F$ values; $F_{lamp}$) are calculated with these lamp observations. Such $\Delta F_{lamp}$ values ($F_{lamp}$ at calibration subtract $F_{lamp}$ at measurement) are used to adjust the instrument response $F$ values (i.e., $F_{adjusted}$ = $F$ - $\Delta F_{lamp}$). Such "SL correction" is included in both ECCC and AEMET Brewer algorithms (e.g., Fioletov et al., 2005; Savastiouk, 2006; Redondas et al., 2018). This means that the reported TCO data used in this work have already accounted for such SL correction. The instrument responses ($F$; e.g., see Eqns 3– 5 in the revised manuscript) used in the Langly fittings also included the SL correction. Note that, this is equivalent to the expression of "adjusted ETC" for the SL test (Fioletov et al., 2005; Redondas et al., 2018).

In short, the SL test results are already included in this work; SL test can be used to monitor the instrumental stability, which typically has small variations due to short-term instrumental changes (e.g., due to the instrument's ambient temperature changes and other factors, see Fig. R7 as an example of SL correction changes for Brewer #119). However, the SL test results should be interpreted carefully, as the values depend on many factors that do not only come from instrumental issues.

[Figure]

**Figure R7. SL correction values from Brewer #119 at MLO from 2021 to 2022.**

We included some of this information in the revised manuscript.

> *Here R6 is a measurement-derived double ratio in the actual Brewer processing algorithm, corresponding to the measured slant column ozone (e.g., Savastiouk, 2006; Zhao et al., 2021).* *Note that in typical conditions (e.g., $\Delta\alpha = 0.34$ and $\mu = 2$), 5 R6 units is equal to 0.74 DU or about 0.25% for a typical ozone value of 300 DU. Also, in this work, the so-called SL (standard lamp) corrections are implemented in the TCO calculation (Fioletov et al., 2005; Savastiouk, 2006; Redondas et al., 2018). The SL corrections are based on internal lamp tests and they compensate for changes in the instrument characteristics that lead to changes in the instrument's ETC and the Langley fitting results (i.e., the instrument responses (F values) used in Langley fits, have the SL correction included). In general, Fig. 1 shows that the selection of fitting equations will not impact the agreement of the final averaged ETC, as long as an adequate number of individual ETCs are included.*

3. "short-term ozone field": It took me a while to decipher its meaning, perhaps it is a modelling term. I would recommend clarifying it when the term first appears on page 1.

Done. We have modified the sentence in the abstract (on page 1).

> *In practice, these two calibration methods have different physical requirements, e.g., the PCM requires* *a stable ozone field in the short-term (i.e., half-day),* *while CTM would benefit from larger changes in slant ozone conditions for the calibration periods.*

**Technical Corrections and Suggestions:**

As Reviewer #1 already mentioned, "site-specific" is more appropriate than "site-specified", which appears in many places in the manuscript.

Done.

P.5, Line 142: "are can be found" -> are found OR can be found

Done.

> *More details about Mark II and III measurements and other characteristics*  *can be found in Zhao et al. (2021).*

Fig. 2. Among the nice figures, this is probably the only one I find difficult to decipher. It is an important figure, perhaps this could be improved. The panels are too small. Also, I would suggest to avoid using red and green markers in the same figure.

Done. We have modified the figure as suggested. The figure is laid out such that each colum represents a group of instruments at one site (e.g., the first column is for instruments at Arosa-Davos). If needed, maybe

the text editor could help us to rotate the figure 90 degrees to make it a full-page figure.

[Figure]

**Figure 2. Long-term Langley fits of Brewers from Arosa/Davos (Brewers #040, #072, #156), Davos (Brewer #163), Izaña (#157, #183, #185), MLO (#119), and Toronto (#145, #187, #191). Gray dots are fitted ETCs for individual half-days, red lines are yearly mean values of these individual ETCs (error bars are 1σ values), and blue lines are ETC values obtained from instrument calibration files (ICF; validation periods are indicated as vertical black dash lines).**

P. 13, Lines 347-348: This (second to the last) sentence needs to be revised or rephrased.

Done.

> *For low-latitude sites (Izaña and MLO), the individual ETCs (gray dots) have less variability and are more closely distributed around the yearly means (red lines) and ICF ETC (blue lines) values.*

P. 23, Line 599: "… the Brewer spectrophotometer taking the most accurate TCO observations among ground-based instruments …". This claim seems like the Brewer has been validated against a "true" measurement of TCO, and then compared with other ground-based instruments, which are also validated against a "true" TCO. If so, please provide references.

Thanks for pointing this out. We agree with the referee that there is no instrument that has been compared with a "true" TCO. We have modified the sentence.

*This is simply due to the Brewer spectrophotometer taking the most reliable TCO observations among ground-based instruments (precision within 1%, corresponding to about 3–4 DU in typical TCO conditions).*

References: Multiple papers of some authors are not in chronological order, e.g. papers by Kerr et al.

Done.